# Plane Section Curves on Surfaces of NCP Functions

**Shun-Wei Li, Yu-Lin Chang *** and **Jein-Shan Chen**

Department of Mathematics, National Taiwan Normal University, Taipei 11677, Taiwan;
wonderful9568509@gmail.com (S.-W.L.); jschen@math.ntnu.edu.tw (J.-S.C.)
* Correspondence: ylchang@math.ntnu.edu.tw

**Abstract:** The goal of this paper is to investigate the curves intersected by a vertical plane with the surfaces based on certain NCP functions. The convexity and differentiability of these curves are studied as well. In most cases, the inflection points of the curves cannot be expressed exactly. Therefore, we instead estimate the interval where the curves are convex under this situation. Then, with the help of differentiability and convexity, we obtain the local minimum or maximum of the curves accordingly. The study of these curves is very useful to binary quadratic programming.

**Keywords:** NCP functions; section curves; convexity; differentiability; binary quadratic programming

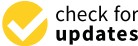



## 1. Introduction

The nonlinear complementarity problem (NCP) is finding a vector $x \in \mathbb{R}^n$ such that

$$x \geq 0, \quad F(x) \geq 0 \quad \text{and} \quad \langle x, F(x) \rangle = 0,$$

where $\langle \cdot, \cdot \rangle$ is the Euclidean inner product and $F$ is a function from $\mathbb{R}^n$ to $\mathbb{R}^n$. Since a few decades ago, the NCP has attracted significant attention due to its various applications in areas such as economics, engineering, and information engineering [1]. There are many methods proposed for solving the NCP. One popular approach is to reformulate the NCP as a system of nonlinear equations, whereas the other approach is to recast the NCP as an unconstrained minimization problem. Both methods rely on the so-called NCP function. A function $\phi : \mathbb{R}^2 \to \mathbb{R}$ is said to be an NCP function if it satisfies

$$\phi(a,b) = 0 \quad \Longleftrightarrow \quad a \geq 0, \quad b \geq 0 \quad \text{and} \quad ab = 0.$$

In light of the NCP function, one can define the vector-valued function $\Phi_F(x) : \mathbb{R}^n \to \mathbb{R}^n$ by

$$\Phi_F(x) := \begin{pmatrix} \phi(x_1, F_1(x)) \\ \vdots \\ \phi(x_n, F_n(x)) \end{pmatrix},$$

where $F(x) = (F_1(x), \cdots, F_n(x))$ is a mapping from $\mathbb{R}^n$ to $\mathbb{R}^n$. Consequently, solving the NCP is equivalent to solving a system of equation $\Phi_F(x) = 0$. In particular, it also induces a merit function of the NCP which is given by

$$\min_{x \in \mathbb{R}^n} \Psi_F(x) := \frac{1}{2} \|\Phi_F(x)\|^2.$$

It is clear that the global minimizer of $\Psi_F(x)$ is the solution to the NCP. During the past few decades, several NCP functions have been discovered [2–7]. A well-known NCP function is the Fischer–Burmeister function [8,9] $\phi_{\text{FB}} : \mathbb{R}^2 \to \mathbb{R}$, defined as

$$\phi_{\text{FB}}(a,b) = ||(a,b)|| - (a+b),$$

where $||(a,b)|| = \sqrt{a^2 + b^2}$. In [10], Tseng did an extension of the Fischer–Burmeister function, in which a 2-norm is relaxed to a general $p$-norm. In other words, the so-called generalized FB function is defined by

$$\phi_{\text{FB}}^p(a,b) = ||(a,b)||_p - (a+b), \tag{1}$$

where $||(a,b)||_p = \sqrt[p]{|a|^p + |b|^p}$ and $p > 1$. Similarly, it induces a merit function $\psi_{\text{FB}}^p : \mathbb{R}^2 \to \mathbb{R}_+$ given by

$$\psi_{\text{FB}}^p(a,b) = \frac{1}{2}|\phi_{\text{FB}}^p(a,b)|^2 \tag{2}$$

where $p > 1$.

Another popular NCP function is the natural residual function [4], $\phi_{\text{NR}} : \mathbb{R} \to \mathbb{R}$ given by

$$\phi_{\text{NR}}(a,b) = a - (a-b)_+.$$

Is there a similar extension for the natural residual NCP function? Wu, Ko and Chen answered this question in [4]. The extension is kind of discrete generalization because they defined the function $\phi_{\text{NR}}^p : \mathbb{R}^2 \to \mathbb{R}$ by

$$\phi_{\text{NR}}^p(a,b) = a^p - (a-b)_+^p, \tag{3}$$

where $p > 1$ and $p$ is an odd integer. Recently, the idea of discrete generalization of natural residual function has beei applied to construct discrete Fischer–Burmeister functions. More specifically, $\phi_{\text{D-FB}}^p : \mathbb{R}^2 \to \mathbb{R}$ is defined by

$$\phi_{\text{D-FB}}^p(a,b) = (\sqrt{a^2 + b^2})^p - (a+b)^p, \tag{4}$$

where $p > 1$ and $p$ is an odd integer. If $p = 1$, then it is exactly the classical Fischer–Burmeister function (see [4,11]). The graph of $\phi_{\text{NR}}^p$ is not symmetric. Is it possible to construct a symmetric natural residual NCP function? Chang, Yang, and Chen answered this question in [2]. Note that the function $\phi_{\text{NR}}^p$ can also be expressed as a piecewise function:

$$\phi_{\text{NR}}^p(a,b) = \begin{cases} a^p - (a-b)^p, & \text{if } a > b, \\ a^p, & \text{if } a \le b, \end{cases}$$

where $p > 1$ and $p$ is an odd integer. They use this expression of $\phi_{\text{NR}}^p$ to modify the part on $a < b$, and achieve symmetrization of $\phi_{\text{S-NR}}^p(a,b)$ as below:

$$\phi_{\text{S-NR}}^p(a,b) = \begin{cases} a^p - (a-b)^p, & \text{if } a > b, \\ a^p = b^p, & \text{if } a = b, \\ b^p - (b-a)^p, & \text{if } a < b, \end{cases} \tag{5}$$

where $p > 1$ and $p$ is an odd integer. Surprisingly, it is still an NCP function.

How about the merit function induced by $\phi_{\text{S-NR}}^p(a,b)$? Observing that the merit function has squared terms, Chang, Yang, and Chen combined $a^p$ and $b^p$ together and constructed $\psi_{\text{S-NR}}^p(a,b)$ as

$$\psi_{\text{S-NR}}^p(a,b) = \begin{cases} a^p b^p - (a-b)^p b^p, & \text{if } a > b, \\ a^p b^p = a^{2p}, & \text{if } a = b, \\ a^p b^p - (b-a)^p a^p, & \text{if } a < b, \end{cases} \tag{6}$$

where $p > 1$ and $p$ is an odd integer.

Recently, more and more NCP functions have been discovered. As mentioned, Wu et al. [4] proposed a discrete type of natural residual function. Regarding this dis-

crete counterpart, Alcantara and Chen [1] consider a continuous type of natural residual function as below:

$$\widetilde{\phi}^p_{\mathrm{NR}}(a,b) = \mathrm{sgn}(a)|a|^p - [(a-b)_+]^p, \tag{7}$$

where $p > 1$ is a real number and

$$\mathrm{sgn}(x) = \begin{cases} 1, & \text{if } x > 0, \\ 0, & \text{if } x = 0, \\ -1, & \text{if } x < 0. \end{cases}$$

The main principle behind their work is described as follows. If $f(\cdot)$ is a bijection mapping and $\phi = \phi_1 - \phi_2$ is a given NCP function, then $f(\phi) = f(\phi_1) - f(\phi_2)$ is also an NCP function. Hence, it can be verified that

$$\widetilde{\phi}^p_{\mathrm{NR}}(a,b) = f(a) - f([a-b]_+)$$

is an NCP function by employing the bijective function $f(t) = \mathrm{sgn}(t)|t|^p$, see [12]. Note that when $p$ is an positive odd integer, it reduces to the discrete type of a natural residual function, that is, $\widetilde{\phi}^p_{\mathrm{NR}}(a,b) = \phi^p_{\mathrm{NR}}(a,b)$.

For further symmetrization, using the above idea in (5) and (6), one can obtain a continuous type of natural residual functions [12]:

$$\widetilde{\phi}^p_{\mathrm{S-NR}}(a,b) = \begin{cases} \mathrm{sgn}(a)|a|^p - (a-b)^p, & \text{if } a \geq b, \\ \mathrm{sgn}(b)|b|^p - (b-a)^p, & \text{if } a < b, \end{cases} \tag{8}$$

and its corresponding merit function

$$\widetilde{\psi}^p_{\mathrm{S-NR}}(a,b) = \begin{cases} \mathrm{sgn}(a)\mathrm{sgn}(b)|a|^p|b|^p - \mathrm{sgn}(b)(a-b)^p|b|^p, & \text{if } a \geq b, \\ \mathrm{sgn}(a)\mathrm{sgn}(b)|a|^p|b|^p - \mathrm{sgn}(a)(b-a)^p|a|^p, & \text{if } a < b, \end{cases} \tag{9}$$

where $p > 0$. Again, when $p$ is an odd integer, we see the beloe relations,

$$\widetilde{\phi}^p_{\mathrm{S-NR}}(a,b) = \phi^p_{\mathrm{S-NR}}(a,b), \quad \text{and} \quad \widetilde{\psi}^p_{\mathrm{S-NR}}(a,b) = \psi^p_{\mathrm{S-NR}}(a,b).$$

The NCP functions can also be constructed by certain invertible functions. What kind of inverse functions can be applied to construct the NCP functions? Lee, Chen, and Hu [6] figured it out in ([6], Proposition 3.8). In particular, let $f : \mathbb{R} \to \mathbb{R}$ be a continuous differentiable function and $g : \mathbb{R} \to \mathbb{R}$ with $g(0) = 1$. They chose functions of $f(t)$ and $g(t)$ satisfying the below conditions to construct new NCP functions:

(i) $f$ is invertible on $[1, \infty)$.

(ii) $(f^{-1})'$ is a strictly monotonically increasing function.

(iii) $g(0) = 1, g(t) \geq 1, \forall t > 0$ and $\frac{-1}{2} < g(t) \leq 1 \; \forall t < 0$.

More specifically, it is shown that the function

$$\phi_{f,g}(a,b) = f(f^{-1}(|a|) + f^{-1}(|b|) - f^{-1}(0)) - (g(b)a + g(a)b)$$

is an NCP function. For example, taking $f(t) = \ln(t)$, we see that $f(t)$ is invertible on $[1, \infty)$ and the inverse function is $f^{-1}(t) = e^t$. It is easy to see that $(f^{-1}(t))' = e^t > 0, \forall t \in \mathbb{R}$. Thus, $f^{-1}$ is strictly monotone increasing on $\mathbb{R}$. For third condition, we take $g(t) = e^t$, which gives $g(t) > 1$ on $(1, \infty)$ and $-\frac{1}{2} < g(t) < 1$ on $(-\infty, 0)$. We list some more examples of $f$ and $g$ as below. Examples of $f(t)$ are

$$f_1(t) = \sqrt{t-1}, \quad f_2(t) = \sqrt[5]{t-1}, \quad f_3(t) = \ln(t),$$

and examples of $g(t)$ are

$$g_1(t) = e^t, \quad g_2(t) = \frac{\sqrt{t^2 + 4} + t}{2}, \quad g_3(t) = \frac{4 - e^{-t}}{1 + 2e^{-t}}.$$

In summary, nine corresponding NCP functions are generated by using the above $f(t)$ and $g(t)$.

$$
\begin{aligned}
\phi_{f_1 g_1}(a, b) &= \sqrt{a^2 + b^2} - e^b a - e^a b. \\
\phi_{f_1 g_2}(a, b) &= \sqrt{a^2 + b^2} - \left( \frac{\sqrt{b^2 + 4} + b}{2} \right) a - \left( \frac{\sqrt{a^2 + 4} + a}{2} \right) b. \\
\phi_{f_1 g_3}(a, b) &= \sqrt{a^2 + b^2} - \left( \frac{4 - e^{-b}}{1 + 2e^{-b}} \right) a - \left( \frac{4 - e^{-a}}{1 + 2e^{-a}} \right) b. \\
\phi_{f_2 g_1}(a, b) &= \sqrt[5]{|a|^5 + |b|^5} - e^b a - e^a b. \\
\phi_{f_2 g_2}(a, b) &= \sqrt[5]{|a|^5 + |b|^5} - \left( \frac{\sqrt{b^2 + 4} + b}{2} \right) a - \left( \frac{\sqrt{a^2 + 4} + a}{2} \right) b. \\
\phi_{f_2 g_3}(a, b) &= \sqrt[5]{|a|^5 + |b|^5} - \left( \frac{4 - e^{-b}}{1 + 2e^{-b}} \right) a - \left( \frac{4 - e^{-a}}{1 + 2e^{-a}} \right) b. \\
\phi_{f_3 g_1}(a, b) &= \ln(e^{|a|} + e^{|b|} - 1) - e^b a - e^a b. \\
\phi_{f_3 g_2}(a, b) &= \ln(e^{|a|} + e^{|b|} - 1) - \left( \frac{\sqrt{b^2 + 4} + b}{2} \right) a - \left( \frac{\sqrt{a^2 + 4} + a}{2} \right) b. \\
\phi_{f_3 g_3}(a, b) &= \ln(e^{|a|} + e^{|b|} - 1) - \left( \frac{4 - e^{-b}}{1 + 2e^{-b}} \right) a - \left( \frac{4 - e^{-a}}{1 + 2e^{-a}} \right) b.
\end{aligned}
\tag{10}
$$

In [13], Tsai et al. discussed the geometry of curves on Fischer–Burmeister function surfaces, which are intersected by the plane $a + b = 2r$ for $r \in \mathbb{R}$. They parametrized the curves by considering $a = r + t$ and $b = r - t$ and defined the vector valued function $\alpha(t) : \mathbb{R} \to \mathbb{R}^3$ and $\beta(t) : \mathbb{R} \to \mathbb{R}^3$ as $\alpha(t) = (r + t, r - t, \phi(r + t, r - t))$ and $\beta(t) = (r + t, r - t, \psi(r + t, r - t))$, respectively. Tsai et al. also found the local maxima and minima and studied the convexity of curves.

In this paper, we follow a similar idea to the one in [13] to investigate the curves, which are the intersection of a vertical plane $a + b = 1$ and surfaces based on NCP functions. We also have to point out that the study on these curves is very useful to binary quadratic programming. See [14] for the details. We parametrize the curves by the vector functions $\tau(x) : \mathbb{R} \to \mathbb{R}^3$ and $\sigma(x) : \mathbb{R} \to \mathbb{R}^3$, where $\tau(x) = (x, 1 - x, \phi(x, 1 - x))$ and $\sigma(x) = (x, 1 - x, \phi(x, 1 - x))$. Then, we explore the behavior of the curves when the value $p$ is perturbed. In addition, we discuss the convexity and local minimum and maximum of curves. Although the inflection points cannot be exactly determined, we can still estimate the interval in which the curves are convex such as in ([14], Proposition 2.1(b)). With the convexity or differentiability of a curve, we discuss the local minimum and maximum.

## 2. Preliminaries

In this section, we review some prerequisite knowledge about the convexity and differentiability of NCP functions which will be applied to investigate the curves. First, it is known that the convexity and differentiability of an NCP function cannot hold simultaneously (see [15]). The convexity of NCP functions has been thoroughly investigated in the literature. We will now quickly recall some results directly.

**Lemma 1** ([3], Property 2.1 and Property 2.2, [2], Proposition 2.2). *Let $\phi_{\mathrm{FB}}^{p}$, $\psi_{\mathrm{FB}}^{p}$ and $\phi_{\mathrm{D-FB}}^{p}$ be defined as in (1), (2) and (4) respectively. Then, the following hold.*

(a) *The function $\phi_{\mathrm{FB}}^{p}(a,b)$ is differentiable everywhere except for the origin, and convex on $\mathbb{R}^{2}$, provided $p > 1$.*

(b) *The function $\psi_{\mathrm{FB}}^{p}(a,b)$ is differentiable everywhere, but neither convex nor concave, provided $p > 1$.*

(c) *The function $\phi_{\mathrm{D-FB}}^{p}(a,b)$ is differentiable everywhere, but neither convex nor concave provided $p > 1$ and is an odd integer.*

**Lemma 2** ([4], Proposition 2.4, [2], Proposition 2.2). *Let $\phi_{\mathrm{NR}}^{p}$, $\phi_{\mathrm{S-NR}}^{p}$, and $\psi_{\mathrm{S-NR}}^{p}$ be defined as in (3), (5) and (6) respectively. Then, when $p > 1$ and is an odd integer, the following hold.*

(a) *The function $\phi_{\mathrm{NR}}^{p}(a,b)$ is differentiable everywhere, but neither convex nor concave.*

(b) *The function $\phi_{\mathrm{S-NR}}^{p}(a,b)$ is differentiable everywhere except for $a = b$. but neither convex nor concave.*

(c) *The function $\psi_{\mathrm{S-NR}}^{p}(a,b)$ is differentiable everywhere, but neither convex nor concave.*

**Lemma 3** ([1], Proposition 2). *Let $\widetilde{\phi}_{\mathrm{NR}}^{p}$, $\widetilde{\phi}_{\mathrm{S-NR}}^{p}$ and $\widetilde{\psi}_{\mathrm{S-NR}}^{p}$ be defined as in (7), (8) and (9) respectively. Then, for $p > 1$, the following hold.*

(a) *The function $\widetilde{\phi}_{\mathrm{NR}}^{p}(a,b)$ is differentiable everywhere, but neither convex nor concave.*

(b) *The function $\widetilde{\phi}_{\mathrm{S-NR}}^{p}(a,b)$ is differentiable everywhere except for $a = b$, but neither convex nor concave.*

(c) *The function $\widetilde{\psi}_{\mathrm{S-NR}}^{p}(a,b)$ is differentiable everywhere, but neither convex nor concave.*

**Proposition 1** ([12], Proposition 2.3). *Suppose that $g$ is strictly increasing on some interval $I = [0, t_0)$. Then, for $p > 1$, the function $\phi_{g}^{p} = \|(a,b)\|_{p} - (g(b)a + g(a)b)$ is an NCP function, but nonconvex.*

We can apply Proposition 1 to check the convexity of NCP functions as in (10). In particular, based on Proposition 1, the following NCP functions are nonconvex and not differentiable at $(0,0)$.

(a) $\phi_{f_1,g_1}(a,b) = \sqrt{a^2 + b^2} - e^b a - e^a b.$

(b) $\phi_{f_1,g_2}(a,b) = \sqrt{a^2 + b^2} - \left( \frac{\sqrt{b^2+4}+b}{2} \right) a - \left( \frac{\sqrt{a^2+4}+a}{2} \right) b.$

(c) $\phi_{f_1,g_3}(a,b) = \sqrt{a^2 + b^2} - \left( \frac{4-e^{-b}}{1+2e^{-b}} \right) a - \left( \frac{4-e^{-a}}{1+2e^{-a}} \right) b.$

(d) $\phi_{f_2,g_1}(a,b) = \sqrt[5]{|a|^5 + |b|^5} - e^b a - e^a b.$

(e) $\phi_{f_2,g_2}(a,b) = \sqrt[5]{|a|^5 + |b|^5} - \left( \frac{\sqrt{b^2+4}+b}{2} \right) a - \left( \frac{\sqrt{a^2+4}+a}{2} \right) b.$

(f) $\phi_{f_2,g_3}(a,b) = \sqrt[5]{|a|^5 + |b|^5} - \left( \frac{4-e^{-b}}{1+2e^{-b}} \right) a - \left( \frac{4-e^{-a}}{1+2e^{-a}} \right) b.$

Moreover, the below NCP functions are nonconvex as well.

(g) $\phi_{f_3,g_1}(a,b) = \ln(e^{|a|} + e^{|b|} - 1) - e^b a - e^a b.$

(h) $\phi_{f_3,g_2}(a,b) = \ln(e^{|a|} + e^{|b|} - 1) - \left( \frac{\sqrt{b^2+4}+b}{2} \right) a - \left( \frac{\sqrt{a^2+4}+a}{2} \right) b.$

(i) $\phi_{f_3,g_3}(a,b) = \ln(e^{|a|} + e^{|b|} - 1) - \left( \frac{4-e^{-b}}{1+2e^{-b}} \right) a - \left( \frac{4-e^{-a}}{1+2e^{-a}} \right) b.$

## 3. The Differentiability of the Curves

In this section, we investigate the differentiability of the curves, which are the intersection of surfaces of NCP functions $\phi(a,b)$, (or merit functions $\psi(a,b)$) with the vertical plane $a + b = 1$. To proceed, we set $a = x$ and $b = 1 - x$. Then, the curves are parameterized as

$$\tau(x) = \phi(x, 1 - x) \quad \text{and} \quad \sigma(x) = \psi(x, 1 - x).$$

From the aforementioned NCP functions in Section 2, the parametrized curves are listed as below:

$$\tau_{\text{FB}}^p(x) = \sqrt[p]{|x|^p + |1-x|^p} - 1. \tag{11}$$

$$\sigma_{\text{FB}}^p(x) = \frac{1}{2}|\tau_{\text{FB}}^p(x)|^2. \tag{12}$$

$$\tau_{\text{D-FB}}^p(x) = \left(\sqrt{x^2 + (1-x)^2}\right)^p - 1. \tag{13}$$

$$\tau_{\text{NR}}^p(x) = x^p - (2x-1)_+^p. \tag{14}$$

$$\tau_{\text{S-NR}}^p(x) = \begin{cases} x^p - (2x-1)^p, & \text{if } x > \frac{1}{2}, \\ (\frac{1}{2})^p, & \text{if } x = \frac{1}{2}, \\ (1-x)^p - (1-2x)^p, & \text{if } x < \frac{1}{2}. \end{cases} \tag{15}$$

$$\sigma_{\text{S-NR}}^p(x) = \begin{cases} x^p(1-x)^p - (2x-1)^p(1-x)^p, & \text{if } x > \frac{1}{2}, \\ (\frac{1}{2})^{2p}, & \text{if } x = \frac{1}{2}, \\ x^p(1-x)^p - x^p(1-2x)^p, & \text{if } x < \frac{1}{2}. \end{cases} \tag{16}$$

$$\widetilde{\tau}_{\text{NR}}^p(x) = \text{sgn}(x)|x|^p - [(2x-1)_+]^p. \tag{17}$$

$$\widetilde{\tau}_{\text{S-NR}}^p(x) = \begin{cases} \text{sgn}(x)|x|^p - (2x-1)^p, & \text{if } x \geq \frac{1}{2}, \\ \text{sgn}(1-x)|1-x|^p - (1-2x)^p, & \text{if } x < \frac{1}{2}. \end{cases} \tag{18}$$

$$\widetilde{\sigma}_{\text{S-NR}}^p(x) = \begin{cases} \text{sgn}(x)\text{sgn}(1-x)|x|^p|1-x|^p - \text{sgn}(1-x)(2x-1)^p|1-x|^p, & \text{if } x \geq \frac{1}{2}, \\ \text{sgn}(x)\text{sgn}(1-x)|1-x|^p|x|^p - \text{sgn}(x)(1-2x)^p|x|^p, & \text{if } x < \frac{1}{2}. \end{cases} \tag{19}$$

$$\tau_{f_1,g_1}(x) = \sqrt{x^2 + (1-x)^2} - e^{(1-x)}x - e^x(1-x). \tag{20}$$

$$\tau_{f_1,g_2}(x) = \sqrt{x^2 + (1-x)^2} - \left(\frac{\sqrt{(1-x)^2+4}+(1-x)}{2}\right)x - \left(\frac{\sqrt{x^2+4}+x}{2}\right)(1-x). \tag{21}$$

$$\tau_{f_1,g_3}(x) = \sqrt{x^2 + (1-x)^2} - \left(\frac{4-e^{-(1-x)}}{1+2e^{-(1-x)}}\right)x - \left(\frac{4-e^{-x}}{1+2e^{-x}}\right)(1-x). \tag{22}$$

$$\tau_{f_2,g_1}(x) = \sqrt[5]{|x|^5 + |1-x|^5} - e^{(1-x)}x - e^x(1-x). \tag{23}$$

$$\tau_{f_2,g_2}(x) = \sqrt[5]{|x|^5 + |1-x|^5} - \left(\frac{\sqrt{(1-x)^2+4}+(1-x)}{2}\right)x - \left(\frac{\sqrt{x^2+4}+x}{2}\right)(1-x). \tag{24}$$

$$\tau_{f_2,g_3}(x) = \sqrt[5]{|x|^5 + |1-x|^5} - \left(\frac{4-e^{-(1-x)}}{1+2e^{-(1-x)}}\right)x - \left(\frac{4-e^{-x}}{1+2e^{-x}}\right)(1-x). \tag{25}$$

$$\tau_{f_3,g_1}(x) = \ln\left(e^{|x|} + e^{|1-x|} - 1\right) - e^{(1-x)}x - e^x(1-x). \tag{26}$$

$$\tau_{f_3,g_2}(x) = \ln\left(e^{|x|} + e^{|1-x|} - 1\right) - \left(\frac{\sqrt{(1-x)^2+4}+(1-x)}{2}\right)x - \left(\frac{\sqrt{x^2+4}+x}{2}\right)(1-x). \tag{27}$$

$$\tau_{f_3,g_3}(x) = \ln(e^{|x|} + e^{|1-x|} - 1) - \left(\frac{4 - e^{-(1-x)}}{1 + 2e^{-(1-x)}}\right)x - \left(\frac{4 - e^{-x}}{1 + 2e^{-x}}\right)(1 - x). \quad (28)$$

**Proposition 2.** *Let $\tau_{\mathrm{FB}}^p$, $\sigma_{\mathrm{FB}}^p$ and $\tau_{\mathrm{D-FB}}^p$ be defined as (11), (12) and (13) respectively. Then, the following hold.*

(a) *For $p > 1$, the function $\tau_{\mathrm{FB}}^p(\cdot)$ is differentiable on $\mathbb{R}$.*

(b) *For $p > 1$, the function $\sigma_{\mathrm{FB}}^p(\cdot)$ is differentiable on $\mathbb{R}$.*

(c) *For all odd integers, the function $\tau_{\mathrm{D-FB}}^p(\cdot)$ is differentiable on $\mathbb{R}$.*

**Proof.** The results follow immediately from Lemma 1. □

**Proposition 3.** *Let $\tau_{\mathrm{NR}}^p(x)$, $\tau_{\mathrm{S-NR}}^p$ and $\sigma_{\mathrm{S-NR}}^p$ be defined as in (14), (15) and (16), respectively. Then, for $p > 1$ and $p$ is an odd integer, the following hold.*

(a) *The function $\tau_{\mathrm{NR}}^p(\cdot)$ is differentiable on $\mathbb{R}$;*

(b) *The function $\tau_{\mathrm{S-NR}}^p(\cdot)$ is not differentiable at $x = \frac{1}{2}$;*

(c) *The function $\sigma_{\mathrm{S-NR}}^p(\cdot)$ is differentiable on $\mathbb{R}$.*

**Proof.** The results are immediate consequences of Lemma 2. □

**Proposition 4.** *Let $\widetilde{\tau}_{\mathrm{NR}}^p$, $\widetilde{\tau}_{\mathrm{S-NR}}^p$, and $\widetilde{\sigma}_{\mathrm{S-NR}}^p$ be defined as in (17), (18) and (19), respectively. Then, for $p > 1$, the following hold.*

(a) *The function $\widetilde{\tau}_{\mathrm{NR}}^p(\cdot)$ is differentiable on $\mathbb{R}$.*

(b) *The function $\widetilde{\tau}_{\mathrm{S-NR}}^p(\cdot)$ is not differentiable at $x = \frac{1}{2}$.*

(c) *The function $\widetilde{\sigma}_{\mathrm{S-NR}}^p(\cdot)$ is differentiable on $\mathbb{R}$.*

**Proof.** The results follow from Lemma 3 directly. □

**Proposition 5.** *Let $\tau_{f_i,g_j}$ be defined as in (20)–(28) where $i = 1, 2, 3$ and $j = 1, 2, 3$. Then, the following hold.*

(a) *For $i = 1, 2$ and $j = 1, 2, 3$, the function $\tau_{f_i,g_j}(\cdot)$ is differentiable on $\mathbb{R}$.*

(b) *For $j = 1, 2, 3$, The function $\tau_{f_3,g_j}(\cdot)$ is not differentiable at $x = 0$ or $x = 1$.*

**Proof.** (a) Based on Proposition 2(a), the function $\tau_{\mathrm{FB}}^p(x)$ is differentiable on $\mathbb{R}$. In addition, we know that the exponential function and $\sqrt{(1 - x)^2 + 4}$ are differentiable on $\mathbb{R}$. Therefore, $\tau_{f_i,g_j}(x)$ is differentiable on $\mathbb{R}$.

(b) Let $h(x) = \ln(e^{|x|} + e^{|1-x|} - 1)$, which says $h'(x) = \frac{\frac{x}{|x|}e^{|x|} - \frac{(1-x)}{|1-x|}e^{|1-x|}}{e^{|x|} + e^{|1-x|} - 1}$. For $x > 0$, the right derivative at $x = 0$ is $h'(0_+) = \frac{1-e}{e}$. For $x < 0$, the left derivative at $x = 0$ is $h'(0_-) = \frac{-1-e}{e}$. Then, it is clear that $h'(0_+) \neq h'(0_-)$, hence $h(\cdot)$ is not differentiable at $x = 0$. Similarly, it is easy to check the non-differentiability at $x = 1$. To summarize, the function $\tau_{f_3,g_j}(x)$ is not differentiable at $x = 0$ or $x = 1$. □

## 4. The Convexity of the Curves

In Section 2, we discussed the convexity of NCP functions. It naturally leads to the convexity of the curves. Although we cannot find the inflection points one by one, we focus on estimating the interval where the curves are convex. In addition, with different $p$, the geometric structure of the curves will be changed. The following lemma will be employed to check the convexity.

**Lemma 4.** (a)  *If $g(x)$ and $h(x)$ are convex on an interval, then $g(x) + h(x)$ is also convex on the interval.*

(b)  *Let $g(x) : \mathbb{R}^n \to (-\infty, \infty)$ be a convex function and let $h(x) : g(\mathbb{R}^n) \to \mathbb{R}$ be a nondecreasing convex function. Then $f(x) = h(g(x))$ is convex on $\mathbb{R}^n$.*

**Proof.** These are very basic materials which are also well known, see [16].  □

**Proposition 6.** *Let $\tau_{\mathrm{FB}}^p$ and $\tau_{\mathrm{D-FB}}^p$ be defined as in (11) and (13), respectively. Then, the following hold. See Figure 1.*

(a)  *For $p > 1$, the function $\tau_{\mathrm{FB}}^p(\cdot)$ is convex on $\mathbb{R}$.*

(b)  *When $p$ is an odd integer, the function $\tau_{\mathrm{D-FB}}^p(\cdot)$ is convex on $\mathbb{R}$.*

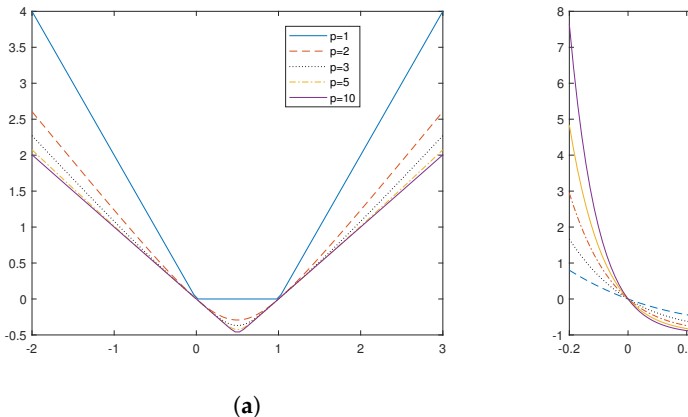

(**a**)       (**b**)

**Figure 1.** Graph of $\tau_{\mathrm{FB}}^p(x)$ and $\tau_{\mathrm{D-FB}}^p(x)$ with different $p$. (**a**) Graph of $\tau_{\mathrm{FB}}^p(x)$ with different values of $p$; (**b**) Graph of $\tau_{\mathrm{D-FB}}^p(x)$ with different values of $p$.

**Proof.** (a) First, as indicated in (11), $\tau_{\mathrm{FB}}^p(x) = \sqrt[p]{|x|^p + |1-x|^p} - 1$. Since the curve $\tau_{\mathrm{FB}}^p(x)$ is the section of a plane with the surface of the function $\phi_{\mathrm{FB}}^p(a, b)$, which is convex on $\mathbb{R}^2$ according to Lemma 1(a). $\tau_{\mathrm{FB}}^p(x)$ is convex on $\mathbb{R}$.

(b) As shown in (13), $\tau_{\mathrm{D-FB}}^p(x) = \left(\sqrt{x^2 + (1-x)^2}\right)^p - 1$, where $p$ is an odd integer. Let $g(x) := \sqrt{x^2 + (1-x)^2}$ and $h(x) := x^p - 1$. It is clear that $h(x)$ is nondecreasing and convex; moreover, $g(x)$ is positive and convex. Then, according to Lemma 4(b), $\tau_{\mathrm{D-FB}}^p(\cdot)$ is convex on $\mathbb{R}$.  □

**Proposition 7.** *Let $\sigma_{\mathrm{FB}}^p$ be defined as in (12). Then, for any $p > 1$, the function $\sigma_{\mathrm{FB}}^p(\cdot)$ is convex on $(-\infty, 0)$ and $(1, \infty)$. See Figure 2.*

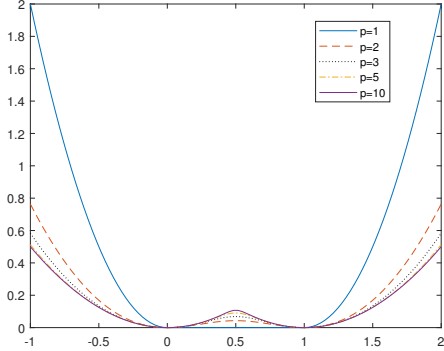

**Figure 2.** Graphs of $\sigma_{\mathrm{FB}}^p$ with different values of $p$.

**Proof.** As given in (12), $\sigma_{\text{FB}}^p(x) = \frac{1}{2}\left(\tau_{FB}^p(x)\right)^2$. Let $g(x) := \tau_{FB}^p(x)$ and $h(x) := \frac{1}{2}x^2$. It is clear that $h(x)$ is nondecreasing and convex on $(0, \infty)$. Furthermore, $g(x)$ is convex and positive on $(1, \infty)$. Hence, according to Lemma 4(b), $\sigma_{\text{FB}}^p(x)$ is convex on $(1, \infty)$. In addition, due to symmetry, $\sigma_{\text{FB}}^p(x)$ is also convex on $(-\infty, 0)$. ☐

**Remark 1.** (i)　*Set $p = 2$. The second derivative of $\sigma_{\text{FB}}^2(x) = \frac{1}{2}|\tau_{\text{FB}}^2(x)|^2$ gives*

$$\left(\sigma_{\text{FB}}^2\right)''(x) = 2 - \frac{1}{(2x^2 - 2x + 1)^{\frac{3}{2}}}.$$

*From this, we know that $a_\pm = \frac{1}{2}\left(1 \pm \sqrt{2^{\left(\frac{1}{3}\right)} - 1}\right)$ are two inflection points of the function $\sigma_{\text{FB}}^p(x)$. Hence, the function $\sigma_{\text{FB}}^p(x)$ is convex on the intervals $(-\infty, a_-)$ and $(a_+, \infty)$. For a general $p > 1$, we have difficulty in determining their infection points. However, let us study their behavior when $p$ goes to $\infty$ on the interval $(0, 1)$. When $1/2 < x < 1$, we have $|x| > |1 - x|$. Hence, the function $\sigma_{\text{FB}}^p(x)$ approaches $\frac{1}{2}(x - 1)^2$ as $p$ goes to $\infty$. Similarly, provided $0 < x < 1/2$, the function $\sigma_{\text{FB}}^p(x)$ approaches $\frac{1}{2}x^2$ as $p$ goes to $\infty$. Note also that $\sigma_{\text{FB}}^p(\frac{1}{2})$ approaches $\frac{1}{8}$ as $p$ goes to $\infty$.*

(ii)　*We also examine the behavior of the second derivative of the function $\sigma_{\text{FB}}^p(x)$ at the point $0.55$ which is near $\frac{1}{2}$. We present the numerical results in Figure 3. Observe that their inflection points $a_\pm^p$ approaches $1/2$, and also that $(\sigma_{\text{FB}}^p)''(0.55)$ approaches $1$ as $p$ goes to $\infty$.*

According to Remark 1 and Figure 3, we make a conjecture here.

**Conjecture 1.** *Let $\sigma_{\text{FB}}^p$ be defined as in (12). Then, for any $p > 1$, the function $\sigma_{\text{FB}}^p(\cdot)$ has two inflection points $0 < a_-^p < \frac{1}{2} < a_+^p < 1$, and both approach $\frac{1}{2}$ as $p$ goes to $\infty$.*

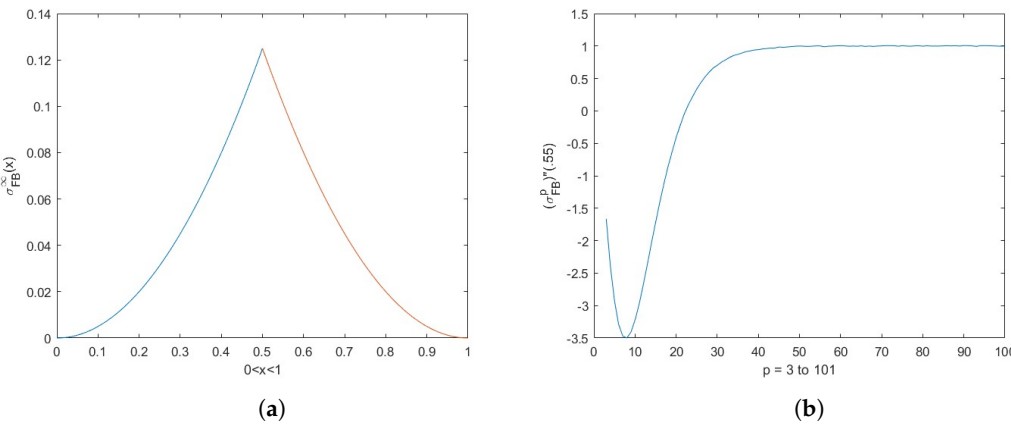

(a)　　　　　　　　　　　　　　　　　　　　　(b)

**Figure 3.** Graphic evidence regarding Remark 1 and Conjecture 1. (**a**) Graphs of $\sigma_{\text{FB}}^\infty(x)$ when $0 < x < 1$; (**b**) Graphs of $(\sigma_{\text{FB}}^p)''(0.55)$ for different $p$.

**Proposition 8.** *Let $\tau_{\text{NR}}^p$ and $\tau_{S-NR}^p$ be defined as in (14) and (15), respectively. Then, when $p$ is an odd integer, the following hold. See Figure 4.*

(a)　*The function $\tau_{\text{NR}}^p(\cdot)$ is convex on $(0, \frac{4}{7})$.*

(b)　*The function $\tau_{S-NR}^p(\cdot)$ is convex on $(\frac{3}{7}, \frac{4}{7})$.*

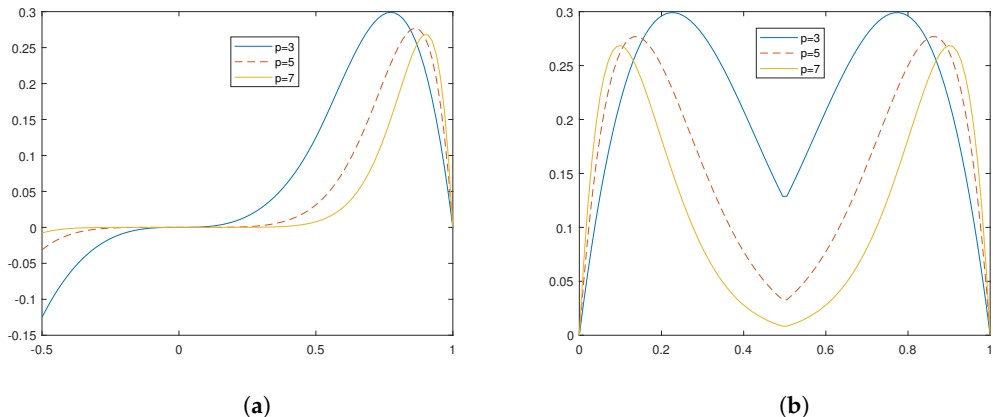

**Figure 4.** Graphs of $\tau_{\text{NR}}^p$ and $\tau_{\text{S-NR}}^p$ with different values of $p$. (**a**) Graphs of $\tau_{\text{NR}}^p$ with different values of $p$; (**b**) Graphs of $\tau_{\text{S-NR}}^p$ with different values of $p$.

**Proof.** (a) As given in (14), $\tau_{\text{NR}}^p(x) = x^p - (2x-1)_+^p$, which says

$$\left(\tau_{\text{NR}}^p\right)'(x) = p\left(x^{(p-1)} - \left[\frac{(2x-1) + |2x-1|}{2}\right]^{(p-1)}(1 + \text{sgn}(2x-1))\right),$$

$$\left(\tau_{\text{NR}}^p\right)''(x) = p(p-1)\left(x^{(p-2)} - \left[\frac{(2x-1) + |2x-1|}{2}\right]^{(p-2)}(1 + \text{sgn}(2x-1))^2\right).$$

To proceed, we discuss three subcases:

Case (i): On the interval $(0, \frac{1}{2})$, we have $(1 + \text{sgn}(2x-1))^2 = 0$, which says $\left(\tau_{\text{NR}}^p\right)''(x)$ $= p(p-1)x^{(p-2)} > 0$.

Case (ii): At the points $a = \frac{1}{2}$, we have $\left(\tau_{\text{NR}}^p\right)''(\frac{1}{2}) = p(p-1)(\frac{1}{2})^{(p-2)} > 0$ as well.

Case (iii): On the interval $(\frac{1}{2}, \frac{4}{7})$, we need to show that $\left(\tau_{\text{NR}}^p\right)''(x) > 0$ over $(\frac{1}{2}, \frac{4}{7})$ for all $p \geq$ 3. Indeed, on the interval $(\frac{1}{2}, \frac{4}{7})$, we have $\left[\frac{(2x-1)+|2x-1|}{2}\right] = 2x-1$ and $(1 + \text{sgn}(2x-1))^2 =$ 4. Define $g_x(p) := x^{(p-2)} - 4(2x-1)^{p-2}$. Then, our goal is to show $g_x(p) > 0$ for all $p \geq 3$ on the interval $(\frac{1}{2}, \frac{4}{7})$. When $p = 3$, we have $g_x(3) = x - 4(2x-1) = -7x + 4 > 0$ on $(\frac{1}{2}, \frac{4}{7})$. In addition, note that $x > 4(2x-1)$ on the same interval. For other $p = 3 + k$ with $k > 0$, we have

$$\begin{aligned} g_x(3+k) &= x^{1+k} - 4(2x-1)^{1+k} \\ &= xx^k - 4(2x-1)^{1+k} \\ &> 4(2x-1)x^k - 4(2x-1)^{1+k} \\ &= 4(2x-1)\left[x^k - (2x-1)^k\right]. \end{aligned}$$

Let $a = 1 - x, b = 2x - 1$. Then, the term $x^k - (2x-1)^k$ in $g_x(3+k)$ is expressed as

$$x^k - (2x-1)^k = (1 - x + 2x - 1)^k - (2x-1)^k = (a+b)^k - b^k.$$

Since, on the interval $(\frac{1}{2}, \frac{4}{7})$ $a$ and $b$ are positive, we conclude that $g_x(3+k) > 0$.

To summarize, on the interval $(\frac{1}{2}, \frac{4}{7})$, the second derivative $\left(\tau_{\text{NR}}^p\right)''(x) > 0$, which means that $\tau_{\text{NR}}^p(x)$ is convex on this interval.

(b) As stated in (15), $\tau^p_{\text{S-NR}}(x) = \begin{cases} x^p - (2x-1)^p & \text{if} \quad x > \frac{1}{2}, \\ (\frac{1}{2})^p & \text{if} \quad x = \frac{1}{2}, \\ (1-x)^p - (1-2x)^p & \text{if} \quad x < \frac{1}{2}. \end{cases}$ For $x > \frac{1}{2}$, similar to part (a), it can be verified that $\tau^p_{\text{NR}}(x)$ is convex. Therefore, $\tau^p_{\text{S-NR}}(x)$ is convex on $(\frac{1}{2}, \frac{4}{7})$. For $x < \frac{1}{2}$, due to symmetry, $\tau^p_{\text{S-NR}}(x)$ is convex on $(\frac{3}{7}, \frac{1}{2})$.

Additionally, note that $\tau^p_{\text{S-NR}}(x)$ is continuous on $(\frac{3}{7}, \frac{4}{7})$, and increasing (decreasing) on the right (left) hand side of the point $a = \frac{1}{2}$, since $\left(\tau^p_{\text{S-NR}}\right)'(1/2_+) = p(1/2)^{p-1} > 0$, $\left(\tau^p_{\text{S-NR}}\right)''(x) > 0$ on the interval $(\frac{1}{2}, \frac{4}{7})$ as well as $\left(\tau^p_{\text{S-NR}}\right)'(1/2_-) = -p(1/2)^{p-1} < 0$, $\left(\tau^p_{\text{S-NR}}\right)''(x) > 0$ on the interval $(\frac{3}{7}, \frac{1}{2})$. Hence, the point $a = \frac{1}{2}$ is the only minimizer on the interval $(\frac{3}{7}, \frac{4}{7})$. In summary, we can conclude that $\tau^p_{\text{S-NR}}(x)$ is convex on the interval $(\frac{3}{7}, \frac{4}{7})$. $\square$

**Proposition 9.** *Let $\sigma^p_{\text{S-NR}}$ be defined as in (16). Then, when $p \geq 3$ and $p$ is an odd integer, the function $\sigma^p_{\text{S-NR}}(x)$ is convex on $(-\infty, 0)$ and $(1, \infty)$. See Figure 5.*

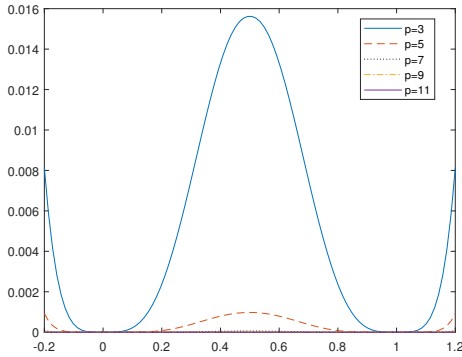

**Figure 5.** Graph of the function $\sigma^p_{\text{S-NR}}$ with different values of $p$.

**Proof.** As indicated in (16), $\sigma^p_{\text{S-NR}}(x) = \begin{cases} x^p(1-x)^p - (2x-1)^p(1-x)^p & \text{if } x > \frac{1}{2}, \\ (\frac{1}{2})^{2p} & \text{if } x = \frac{1}{2}, \\ x^p(1-x)^p - x^p(1-2x)^p & \text{if } x < \frac{1}{2}, \end{cases}$ where $p$ is an odd integer and $p > 1$. Since $\sigma^p_{\text{S-NR}}(x)$ is symmetric about $x = \frac{1}{2}$, we divide it into two cases:

Cases (i): Suppose $x > 1$, the first and second derivative of this function are

$$\left(\sigma^p_{S-NR}\right)'(x) = -p(1-x)^{p-1}[x^p - (2x-1)^p] + (1-x)^p[p(x^{p-1} - 2(2x-1)^{p-1})]$$
$$\left(\sigma^p_{S-NR}\right)''(x) = f_1(x) + f_2(x) + f_3(x)$$

where

$$f_1(x) = (-2p)p[x^{p-1} - 2(2x-1)^{p-1}](1-x)^{p-1} = x^{p-1}[1 - 2(2 - \tfrac{1}{x})^{p-1}](1-x)^{p-1}(-2p)p,$$

$$f_2(x) = p(p-1)[x^{p-2} - 4(2x-1)^{p-2}](1-x)^p = x^{p-2}[1 - 4(2 - \tfrac{1}{x})^{p-1}](1-x)^p(p-1)p,$$

$$f_3(x) = p(p-1)[x^p - (2x-1)^p](1-x)^{p-2} = x^p[1 - (2 - \tfrac{1}{x})^{p-2}](1-x)^{p-2}(p-1)p.$$

Note that $\left(\sigma^p_{S-NR}\right)''(1) = 0$, we want to show that $\left(\sigma^p_{S-NR}\right)''(x)$ is positive for $p > 1$.

Because $x > 1$, we have $1 < 2 - \frac{1}{x}$, which implies $1 - 2(2 - \frac{1}{x})^{p-1} < -1$. Moreover, as we have $(-2p)px^{p-1} < 0$ and $(1-x)^{p-1} > 0$, then $f_1(x) > 0$. Similarly, because $x > 1$,

we have $1 < 2 - \frac{1}{x}$. Hence, $1 - 4(2 - \frac{1}{x})^{p-1} < -3$. Moreover, as we have $(p-1)px^{p-2} > 0$ and $(1-x)^p < 0$, then $f_2(x) > 0$. Finally, because $x > 1$ we have $1 < 2 - \frac{1}{x}$, which gives $1 - (2 - \frac{1}{x})^{p-2} < 0$. Moreover, we have $(p-1)px^p > 0$ and $(1-x)^{p-2} < 0$. Then, it says $f_3(x) > 0$.

To summarize, we have shown $f_1(x) + f_1(x) + f_1(x) > 0$ for $x > 1$, which says $\left(\sigma_{S-NR}^p\right)''(x) > 0$ for $x > 1$. In other words, $\sigma_{S-NR}^p(\cdot)$ is convex on $(1, \infty)$.

Cases (ii): Suppose $x < 0$, since $\sigma_{S-NR}^p(x)$ is symmetric about $x = \frac{1}{2}$. In this case, it is clear that $\sigma_{S-NR}^p(x)$ is convex on $(-\infty, 0)$.

By cases (i) and (ii), we prove that $\sigma_{S-NR}^p(x)$ is convex on $(-\infty, 0)$ and $(1, \infty)$. $\quad\square$

Because $\widetilde{\tau}_{NR}^p$, $\widetilde{\tau}_{S-NR}^p$ and $\widetilde{\sigma}_{S-NR}^p$ are the continuous types of $\tau_{NR}^p$, $\tau_{S-NR}^p$ and $\sigma_{S-NR}^p$, similar to Propositions 8 and 9, we establish the next proposition.

**Proposition 10.** *Let $\widetilde{\tau}_{NR}^p$, $\widetilde{\tau}_{S-NR}^p$ and $\widetilde{\sigma}_{S-NR}^p$ be defined as in (17), (18), and (19), respectively. Then, the following hold. See Figure 6.*

(a) *If $p \geq 3$, then the function $\widetilde{\tau}_{NR}^p(x)$ is convex on $(0, \frac{4}{7})$.*
(b) *If $p \geq 3$, then the function $\widetilde{\tau}_{S-NR}^p(x)$ is convex on $(\frac{3}{7}, \frac{4}{7})$.*
(c) *If $p \geq 3$, then the function $\widetilde{\sigma}_{S-NR}^p(x)$ is convex on $(-\infty, 0)$ and $(1, \infty)$.*

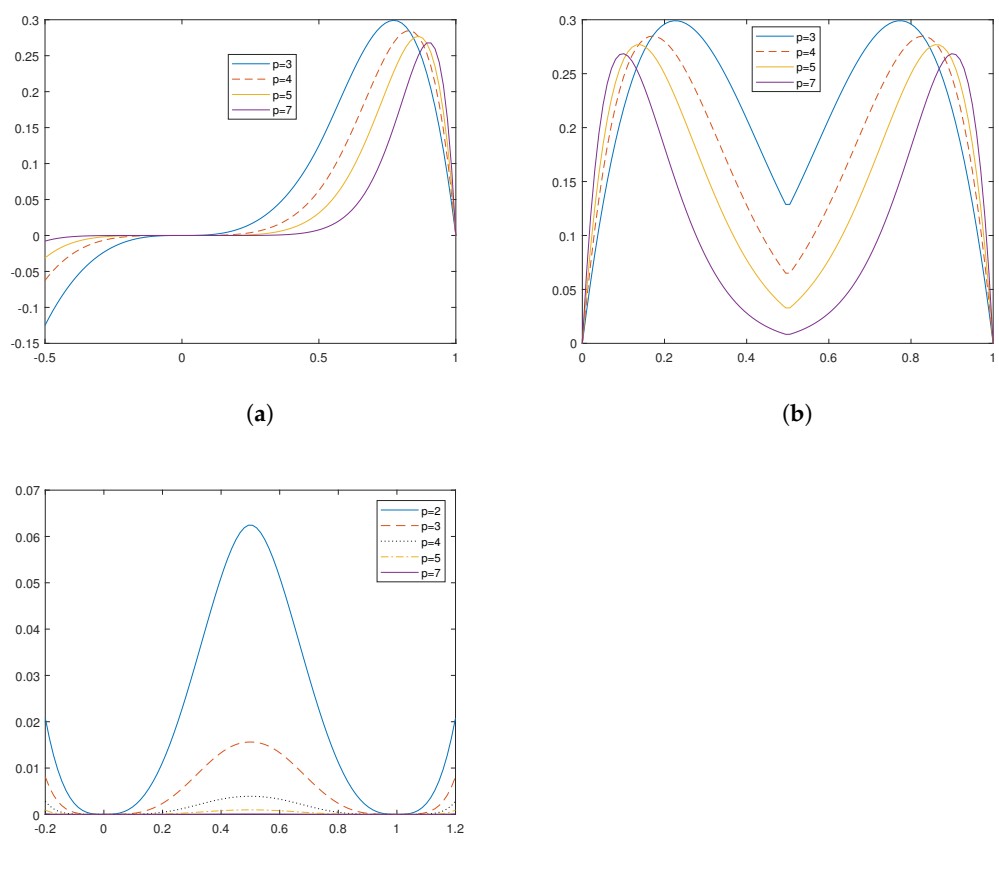

**Figure 6.** Graphs of $\widetilde{\tau}_{NR}^p(x)$, $\widetilde{\tau}_{S-NR}^p(x)$ and $\widetilde{\sigma}_{S-NR}^p$ with different values of $p$. (**a**) Graphs of $\widetilde{\tau}_{NR}^p(x)$ with different values of $p$; (**b**) Graphs of $\widetilde{\tau}_{S-NR}^p(x)$ with different values of $p$; (**c**) Graphs of $\widetilde{\sigma}_{S-NR}^p$ with different values of $p$.

The following proposition is simple but tedious. We list it here for the readers' convenience.

**Proposition 11.** *Let $\tau_{f_i,g_j}$ where $i = 1, 2$ and $j = 1, 2, 3$ be defined from (20)–(28). Then, the following hold.*

(a)  *The function $\tau_{f_i,g_j}(\cdot)$ for $i = 1, 2$ and $j = 1, 2$ is convex on $\mathbb{R}$.*

(b)  *The function $\tau_{f_3,g_j}(\cdot)$ for $j = 1, 2$ is convex on intervals $(-\infty, 0)$, $(0, 1)$ and $(1, \infty)$.*

(c)  *The function $\tau_{f_i,g_3}(\cdot)$ for $i = 1, 2, 3$ has inflection points, and thus is neither convex nor concave on entire $\mathbb{R}$.*

**Proof.** (a) As stated in (20), $\tau_{f_1,g_1}(x) = \sqrt{x^2 + (1-x)^2} - e^{(1-x)}x - e^x(1-x)$. Let $g(x) := \sqrt{x^2 + (1-x)^2}$ and $h(x) := -e^{(1-x)}x - e^x(1-x)$. Because $g(x)$ is convex on $\mathbb{R}$ according to Proposition 6(b), it suffices to show that $h(x)$ is convex. Taking the first and second derivatives of this function give

$$h'(x) = e^{(1-x)}(x-1) + e^x x, \quad \text{and} \quad h''(x) = xe^x + e^x - e^{1-x}x + 2e^{1-x}.$$

In order to verify that $h''(x) > 0$, we divide it into three cases:

Cases (i) : Suppose $x \geq 1/2$. We have $x \geq 1 - x$, hence $e^x \geq e^{1-x}$. Then, we obtain $h''(x) = x(e^x - e^{1-x}) + e^x + 2e^{1-x} > 0$.

Cases (ii): Suppose $0 \leq x \leq 1/2$. We have $2 > x$, hence $2e^{1-x} > xe^{1-x}$. Then, we obtain $h''(x) = (2e^{1-x} - xe^{1-x}) + e^x + xe^x > 0$.

Cases (iii): Suppose $x \leq 0$. We have $x \leq 1 - x$, hence $e^x \leq e^{1-x}$. Then, we obtain $h''(x) = x(e^x - e^{1-x}) + e^x + 2e^{1-x} > 0$.

This shows that $h''(x)$ is always positive, which indicates that $h(x)$ is convex on $\mathbb{R}$. Because $g(x)$ and $h(x)$ are convex on $\mathbb{R}$, according to Lemma 4(a), the function $\tau_{f_1,g_1}(\cdot)$ is convex on $\mathbb{R}$.

As indicated in (21), $\tau_{f_1,g_2}(x) = \sqrt{x^2 + (1-x)^2} - \left( \frac{\sqrt{(1-x)^2+4}+(1-x)}{2} \right)x - \left( \frac{\sqrt{x^2+4}+x}{2} \right)(1-x)$. Let $h(x) := \sqrt{x^2 + (1-x)^2}$ and $g(x) := -\left( \frac{\sqrt{(1-x)^2+4}+(1-x)}{2} \right)x - \left( \frac{\sqrt{x^2+4}+x}{2} \right)(1-x)$. We need to verify that $g(x)$ is convex. Taking the second derivative of $g(x)$ gives

$$g''(x) = \frac{-x^3 + 3x^2 - 9x + 5}{(x^2 - 2x + 5)^{\frac{3}{2}}} + \frac{x^3 + 6x - 2}{(x^2 + 4)^{\frac{3}{2}}} + 2.$$

We want to show that that $g''(x) > 0$. The main principle of this is to check whether the minimum of the second derivative is positive. Taking the third derivative gives

$$g'''(x) = \frac{6(x+4)}{(x^2+4)^{\frac{5}{2}}} + \frac{6(x-5)}{(x^2 - 2x + 5)^{\frac{5}{2}}}$$

The critical numbers of $g''(x)$ are $x \approx \frac{1}{2}, -1.946503$, and $2.946503$. Moreover, $g''(\frac{1}{2}) \approx 2.2568$, and $g''(-1.946503) = g''(2.946503) \approx 1.945045$. The intervals where it is increasing are $(-1.946503, \frac{1}{2})$ and $(2.946503, \infty)$, and the intervals where it is decreasing are $(-\infty, -1.946503)$ and $(\frac{1}{2}, 2.946503)$. Therefore, the local minimum is $1.945045$, and the local maximum is $2.2568$. Furthermore, we also find $\lim_{x \to \pm\infty} g''(x) = 2$. This shows that the global minimum of $g''(x)$ is positive, hence $g''(x) > 0$ on the entire $\mathbb{R}$. This implies that $g(x)$ is convex on $\mathbb{R}$. As $h(x)$ and $g(x)$ are convex on $\mathbb{R}$ according to Lemma 4(a), $\tau_{f_1,g_2}(\cdot)$ is convex on $\mathbb{R}$.

As shown in (23), $\tau_{f_2,g_1}(x) = \sqrt[5]{|x|^5 + |1-x|^5} - e^{(1-x)}x - e^x(1-x)$. As $\sqrt[5]{|x|^5 + |1-x|^5}$ and $-e^{(1-x)}x - e^x(1-x)$ are convex on $\mathbb{R}$ from previous discussions according to Lemma 4(a), $\tau_{f_2,g_1}(x)$ is convex on $\mathbb{R}$.

As given in (24), $\tau_{f_2,g_2}(x) = \sqrt[5]{|x|^5 + |1-x|^5} - \left(\frac{\sqrt{(1-x)^2+4}+(1-x)}{2}\right)x - \left(\frac{\sqrt{x^2+4}+x}{2}\right)(1-x)$. As $\sqrt[5]{|x|^5 + |1-x|^5}$ and $-\left(\frac{\sqrt{(1-x)^2+4}+(1-x)}{2}\right)x - \left(\frac{\sqrt{x^2+4}+x}{2}\right)(1-x)$ are convex on $\mathbb{R}$ from previous work according to Lemma 4(a), $\tau_{f_2,g_2}(x)$ is convex on $\mathbb{R}$.

(b) As shown in (26), $\tau_{f_3,g_1}(x) = \ln(e^{|x|} + e^{|1-x|} - 1) - e^{(1-x)}x - e^x(1-x)$. Let $h(x) := \ln(e^{|x|} + e^{|1-x|} - 1)$ and $g(x) := -e^{(1-x)}x - e^x(1-x)$. As $g(x)$ is convex on $\mathbb{R}$ based on the proof of the case for $\tau_{f_1,g_1}$, the convexity of $h(x)$ is all that remains to determined. Note that $h(x)$ is not differentiable at $x = 0$ and $x = 1$, and we need to discuss three cases:

Cases (i): Suppose $0 < x < 1$. Taking the first derivative and second derivative of $h(x)$ give

$$h'(x) = \frac{\frac{xe^{|x|}}{|x|} - \frac{(1-x)e^{|1-x|}}{|1-x|}}{e^{|x|} + e^{|1-x|} - 1},$$

$$h''(x) = \frac{(e^{|x|} + e^{|1-x|})(e^{|x|} + e^{|1-x|} - 1) - (\frac{x}{|x|}e^{|x|} - \frac{(1-x)}{|1-x|}e^{|1-x|})^2}{(e^{|x|} + e^{|1-x|} - 1)^2}.$$

Since the denominator of $h''(x)$ is positive, we need to check whether the numerator is positive. The numerator is $(e^x + e^{1-x})^2 - (e^x + e^{1-x}) - (e^x - e^{1-x})^2 = 4e - (e^x + e^{1-x})$. For $0 < x < 1$, we have $1 < e^x < e$ and $1 < e^{(1-x)} < e$, which indicates that the numerator is positive. Therefore, we conclude $h''(x) > 0$, and hence $\tau_{f_3,g_1}(\cdot)$ is convex on the interval $(0,1)$.

Cases (ii): Suppose $x > 1$, taking the second derivative of $h(x)$ gives

$$h''(x) = \frac{-\left(e^x + e^{(x-1)}\right)}{\left(e^x + e^{(x-1)} - 1\right)^2} + e^x(x+1) - e^{1-x}(x-2).$$

We want to show that $h''(x) > 0$ for $x > 1$. Taking the third derivative of $h(x)$ yields

$$h'''(x) = \left(\frac{(e^x + e^{x-1} + 1)(e^x + e^{x-1})}{(e^x + e^{x-1} - 1)^3}\right) + (e^{1-x}(x-3) + e^x(x+2)).$$

For the first term of $h'''(x)$, since $e^x + e^{x-1} > e + 1$, the denominator is positive, and hence the first term is positive. For the second term of $h'''(x)$, we have

$$e^{1-x}(x-3) + e^x(x+2) = e^{1-x}x + e^xx + 2e^x - 3e^{(1-x)} = e^{1-x}x + e^xx + 2ee^{x-1} - 3e^{(1-x)}.$$

As $2e > 3$ and $e^{x-1} > e^{1-x}$ when $x > 1$, it is also positive. Therefore, we obtain $h'''(x) > 0$. This shows that $h''(x)$ is increasing. Note also that $h''(1) = 1 + 2e - \frac{1+e}{e^2} > 0$. Then, it follows that $h''(x) > 0$. $\tau_{f_3,g_1}(x)$ is convex on the interval $(1, \infty)$.

Cases (iii): Suppose $x < 0$. As $\tau_{f_3,g_1}(x)$ is symmetric about the point $x = \frac{1}{2}$ according to case (ii), the function $\tau_{f_3,g_1}(\cdot)$ is convex on interval $(-\infty, 0)$.

As indicated in (27),

$$\tau_{f_3,g_2}(x) = \ln(e^{|x|} + e^{|1-x|} - 1) - \left(\frac{\sqrt{(1-x)^2+4}+(1-x)}{2}\right)x - \left(\frac{\sqrt{x^2+4}+x}{2}\right)(1-x).$$

Let $h(x) := \ln(e^{|x|} + e^{|1-x|} - 1)$ and $g(x) := -\left(\frac{\sqrt{(1-x)^2+4}+(1-x)}{2}\right)x - \left(\frac{\sqrt{x^2+4}+x}{2}\right)(1-x)$. $g(x)$ is convex on $\mathbb{R}$ according to the proof of the case for $\tau_{f_1,g_2}$ and $h(x)$ is convex on the intervals $(-\infty, 0)$, $(0,1)$ and $(1, \infty)$ according to previous arguments. Therefore, $\tau_{f_3,g_2}$ is convex on the intervals $(-\infty, 0)$, $(0,1)$, and $(1, \infty)$.

(c) As given in (22), $\tau_{f_1,g_3}(x) = \sqrt{x^2 + (1-x)^2} - (\frac{4 - e^{-(1-x)}}{1 + 2e^{-(1-x)}})x - (\frac{4 - e^{-x}}{1 + 2e^{-x}})(1-x)$. Taking the second derivative of $\tau_{f_1,g_3}(x)$ gives

$$\left(\tau_{f_1,g_3}\right)''(x) \;=\; \frac{1}{(2x^2 - 2x + 1)^{\frac{3}{2}}} + \frac{9}{2}\left(-\frac{e(x-2)}{2e^x + e} - \frac{2e^3 x}{(2e^x + e)^3}\right.$$
$$\left. + \frac{e^2(3x-2)}{(2e^x + e)^2} + \frac{4e^x(x+1) - 2e^{2x}(x-3)}{(e^x + 2)^3}\right).$$

The inflection points are $x \approx -1.986749$, $2.986749$, $-12.999449$, and $13.99944$. Then, the intervals where the curve is convex are $(-1.986749, 2.986749)$, $(-\infty, -12.999449)$ and $(13.999449, \infty)$.

As indicated in (25), we know

$$\tau_{f_2,g_3}(x) = \sqrt[5]{|x|^5 + |1-x|^5} - \left(\frac{4 - e^{-(1-x)}}{1 + 2e^{-(1-x)}}\right)x - \left(\frac{4 - e^{-x}}{1 + 2e^{-x}}\right)(1-x).$$

Similarly, we use the second derivative to find the inflection points. The inflection points are $x \approx 3.005175$, $-2.005175$, $-11.286820$, and $12.286820$. Therefore, the intervals where the curve is convex are $(-2.005175, 3.005175)$, $(12.286820, \infty)$, and $(-\infty, -11.286820)$.

As shown in (28), we know

$$\tau_{f_3,g_3}(x) = \ln(e^{|x|} + e^{|1-x|} - 1) - \left(\frac{4 - e^{-(1-x)}}{1 + 2e^{-(1-x)}}\right)x - \left(\frac{4 - e^{-x}}{1 + 2e^{-x}}\right)(1-x).$$

Similarly, we use the second derivative to find the inflection points. The inflection points are $x \approx -1.904132$ and $2.904132$. Because $\ln(e^{|x|} + e^{|1-x|} - 1)$ is not differentiable at the points 0 and 1, we can only assure that the interval where the curve is convex is $(0, 1)$. $\square$

Recall that a function is called subdifferentiable at $x$ if there exists at least one subgradient at $x$. Although $\tau_{f_3,g_1}(x)$ is not differentiable at the points 0 and 1, with the help of Proposition 11(b), we can still show that it is subdifferentiable thereat.

**Proposition 12.** (a) *The function $\tau_{f_3,g_1}(\cdot)$ is subdifferentiable at the points 0 and 1 and the subdifferential is described by*

$$\partial\tau_{f_3,g_1}(0) \;=\; \left[\frac{(-1-e)}{e} - e, \frac{(1-e)}{e} - e\right],$$
$$\partial\tau_{f_3,g_1}(1) \;=\; \left[\frac{(e-1)}{e} + e, \frac{(e+1)}{e} + e\right].$$

*Moreover, $\tau_{f_3,g_1}(\cdot)$ is convex on $\mathbb{R}$.*

(b) *The function $\tau_{f_3,g_2}(\cdot)$ is subdifferentiable at the points 0 and 1 and the subdifferential is described by*

$$\partial\tau_{f_3,g_2}(0) \;=\; \left[\frac{(-1-e)}{e} - \frac{\sqrt{5}}{2}, \frac{(1-e)}{e} - \frac{\sqrt{5}}{2}\right],$$
$$\partial\tau_{f_3,g_2}(1) \;=\; \left[\frac{(e-1)}{e} + \frac{\sqrt{5}}{2}, \frac{(e+1)}{e} + \frac{\sqrt{5}}{2}\right].$$

*Moreover, $\tau_{f_3,g_2}(\cdot)$ is convex on $\mathbb{R}$.*

**Proof.** (a) Taking the first derivative of $\tau_{f_{3,g_1}}(x)$ gives

$$\left(\tau_{f_{3,g_1}}\right)'(x) = \frac{\frac{xe^{|x|}}{|x|} - \frac{(1-x)e^{|1-x|}}{|1-x|}}{e^{|x|} + e^{|1-x|} - 1} + \left(e^{(1-x)}(x-1) + e^x x\right).$$

The right and left derivatives at the point 0 are $\left(\tau_{f_{3,g_1}}\right)'(0_+) = \frac{1-e}{e} - e$ and $\left(\tau_{f_{3,g_1}}\right)'(0_-)$ $= \frac{-1-e}{e} - e$, respectively. Moreover, we have $\left(\tau_{f_{3,g_1}}\right)'(0_+) > \left(\tau_{f_{3,g_1}}\right)'(0_-)$. Based on the convexity of $\tau_{f_{3,g_1}}(x)$ on $(-\infty, 0)$ from Proposition 11(b), we have

$$\tau_{f_{3,g_1}}(\epsilon_- + h) - \tau_{f_{3,g_1}}(\epsilon_-) \geq \left(\tau_{f_{3,g_1}}\right)'(\epsilon_-)h$$

with small $\epsilon_- < 0$ and $h < 0$. Note here the $\tau_{f_{3,g_1}}(x)$ is a continuous function. Let $\epsilon_- \to 0$. Thus, we have $\tau_{f_{3,g_1}}(h) - \tau_{f_{3,g_1}}(0) \geq \left(\tau_{f_{3,g_1}}\right)'(0_-)h$ for $h < 0$. Similarly, according to the convexity of $\tau_{f_{3,g_1}}(x)$ on $(0, 1)$ from Proposition 11(b), we can obtain that $\tau_{f_{3,g_1}}(h) - \tau_{f_{3,g_1}}(0) \geq \left(\tau_{f_{3,g_1}}\right)'(0_+)h$ where $0 < h < 1$. Therefore, we show that $\tau_{f_{3,g_1}}(x)$ is subdifferentiable at 0, and $\partial\tau_{f_{3,g_1}}(0) = \left[\frac{(-1-e)}{e} - e, \frac{(1-e)}{e} - e\right]$. Moreover based on Lemma 2.13 in [17], $\tau_{f_{3,g_1}}(x)$ is convex on the interval $(-\infty, 1)$, especially at the point 0. Likewise, $\partial\tau_{f_{3,g_1}}(1) = \left[\frac{e-1}{e} + e, \frac{e+1}{e} + e\right]$ and it is convex at the point 1. Hence, $\tau_{f_{3,g_1}}(x)$ is convex on entire $\mathbb{R}$.

(b) Taking the first derivative of $\tau_{f_{3,g_2}}(x)$ yields

$$\left(\tau_{f_{3,g_2}}\right)'(x) = \frac{\frac{xe^{|x|}}{|x|} - \frac{(1-x)e^{|1-x|}}{|1-x|}}{e^{|x|} + e^{|1-x|} - 1}$$
$$+ \frac{1}{2}\left[(1-x)\left(\frac{-x}{\sqrt{x^2+4}} - 1\right) + \sqrt{x^2+4} - \frac{(x-1)x}{\sqrt{x(x-2)+5}} + 3x - \sqrt{x(x-2)+5} - 1\right].$$

The right derivative at the point 0 is $\left(\tau_{f_{3,g_2}}\right)'(0_+) = \left(\frac{1-e}{e}\right) - \frac{\sqrt{5}}{2}$ and the left derivative at the point 0 is $\left(\tau_{f_{3,g_2}}\right)'(0_-) = \frac{-1-e}{e} - \frac{\sqrt{5}}{2}$. Therefore, we obtain

$$\partial\tau_{f_{3,g_2}}(0) = \left[\frac{-1-e}{e} - \frac{\sqrt{5}}{2}, \frac{1-e}{e} - \frac{\sqrt{5}}{2}\right].$$

Similarly, $\partial\tau_{f_{3,g_2}}(1) = \left[\frac{e-1}{e} + \frac{\sqrt{5}}{2}, \frac{e+1}{e} + \frac{\sqrt{5}}{2}\right]$. $\quad\square$

## 5. The Local Minimum and Maximum of the Curves

After discussing the convexity and differentiability, we now work on finding the local minimum or maximum value of the curves. In addition, we shall investigate the convergent behavior of local minimum or maximum values when $p$ becomes very large.

**Proposition 13.** *Let $\tau_{FB}^p$, $\tau_{D-FB}^p$ and $\sigma_{FB}^p$ be defined as in (11), (13), and (12) respectively. Then, the following hold. See Figure 7.*

(a)  *The function $\tau_{FB}^p(x)$ has a local minimum at $x = \frac{1}{2}$ and its local minimum value converges to $-\frac{1}{2}$.*

(b)  *When $p$ is an odd integer, the function $\tau_{D-FB}^p(x)$ has a local minimum at $x = \frac{1}{2}$ and its local minimum value converges to $-1$.*

(c) *The function $\sigma_{\text{FB}}^{p}(x)$ has local minima at $x = 0$ and $1$. Furthermore, it has a local maximum value at $x = \frac{1}{2}$ and its local maximum value converges to $\frac{1}{8}$.*

**Proof.** (a) From (11), we know that

$$\tau_{\text{FB}}^{p}(x) = \sqrt[p]{|x|^{p} + |1 - x|^{p}} - 1$$

where $p > 1$. The first derivative of this function is

$$\left(\tau_{\text{FB}}^{p}\right)'(x) = (|x|^{p} + |1 - x|^{p})^{\frac{1}{p} - 1}[\text{sgn}(x)|x|^{p-1} - |1 - x|^{p-1}\text{sgn}(1 - x)].$$

Note that the first term is positive. We then investigate the second term:

$$\Big[\text{sgn}(x)|x|^{p-1} - |1 - x|^{p-1}\text{sgn}(1 - x)\Big].$$

Case (i): If $x > 1/2$, then $\text{sgn}(x)|x|^{p-1} - |1 - x|^{p-1}\text{sgn}(1 - x) > 0$.

Case (ii): If $x < 1/2$, then $\text{sgn}(x)|x|^{p-1} - |1 - x|^{p-1}\text{sgn}(1 - x) < 0$.

Case (iii): When $x = \frac{1}{2}$, we see that $a = \frac{1}{2}$ is the only root of $\left(\tau_{\text{FB}}^{p}\right)'(x) = 0$. Moreover, $\tau_{\text{FB}}^{p}(x)$ is convex on $\mathbb{R}$, which indicates $a = \frac{1}{2}$ is the only local minimizer and the value is $(\frac{1}{2})(2^{\frac{1}{p}}) - 1$. Furthermore, we observe that the local minimum value converges to $-\frac{1}{2}$ as $p \to \infty$.

(b) From (13), we know that

$$\tau_{\text{D−FB}}^{p}(x) = (\sqrt{x^2 + (1 - x)^2})^{p} - 1$$

where $p > 1$ and $p$ is an odd integer. Taking the first derivative of this function yields

$$\left(\tau_{\text{D−FB}}^{p}\right)'(x) = p(x^2 + (1 - x)^2)^{\frac{p}{2} - 1}(2x - 1).$$

It can be verified that $a = \frac{1}{2}$ is the singular critical point. Note that $\tau_{\text{D−FB}}^{p}(x)$ is convex on $\mathbb{R}$, hence $a = \frac{1}{2}$ is a local minimizer and the value is $\left(\sqrt{2(\frac{1}{2})^2}\right)^{p} - 1$. In addition, the local minimum value converges to $-1$ when $p \to \infty$.

(c) From (12), we know that

$$\sigma_{\text{FB}}^{p}(x) = \frac{1}{2}|\tau_{\text{FB}}^{p}(x)|^2$$

where $p > 1$. Taking the first derivative of this function gives

$$\left(\sigma_{\text{FB}}^{p}\right)'(x) = \tau_{\text{FB}}^{p}(x)\left(\tau_{\text{FB}}^{p}\right)'(x).$$

We want to solve $\left(\sigma_{\text{FB}}^{p}\right)'(x) = 0$, which implies $\tau_{\text{FB}}^{p}(x) = 0$ or $\left(\tau_{\text{FB}}^{p}\right)'(x) = 0$. If $\tau_{\text{FB}}^{p}(x) = 0$, we have $x = 0$ and $x = 1$. If $\left(\tau_{\text{FB}}^{p}\right)'(x) = 0$, we have $x = \frac{1}{2}$. Thus, the critical numbers are $x = 0, \frac{1}{2}, 1$. Note that $0$ and $1$ are the only two roots of $\tau_{\text{FB}}^{p}(x) = 0$ and $\tau_{\text{FB}}^{p}(x)$ is non-negative. Therefore, we see that $x = 0$ and $x = 1$ are local minimizers, and the values are both $0$.

On the other hand, we know that $\tau_{\text{FB}}^{p}(x)$ is decreasing (increasing) on the right (left) hand side of the point $a = \frac{1}{2}$. Hence, the point $a = \frac{1}{2}$ is a local maximizer, and the value is $\frac{1}{2}\left[(2(\frac{1}{2})^{p})^{\frac{1}{p}} - 1\right]^{2}$. This further implies that when $p \to \infty$, the local maximum converges to $\frac{1}{8}$. $\square$

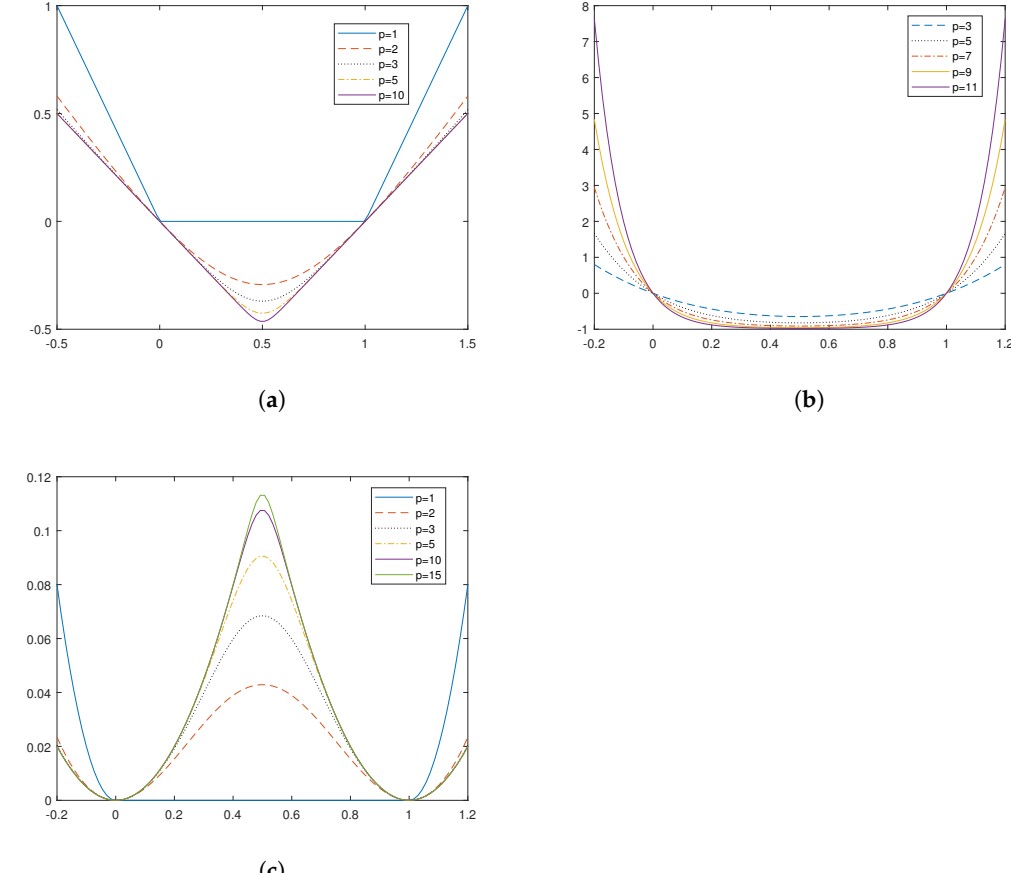

**Figure 7.** Graphs of $\tau_{\text{FB}}^p(x)$, $\tau_{\text{D-FB}}^p(x)$ and $\sigma_{\text{FB}}^p(x)$ with different values of $p$. (**a**) Local minimum of $\tau_{\text{FB}}^p(x)$; (**b**) Local minimum of $\tau_{\text{D-FB}}^p(x)$; (**c**) Local minimum and maximum of $\sigma_{\text{FB}}^p(x)$.

**Proposition 14.** *Let $\tau_{\text{NR}}^p$ be defined as in (14) with odd integer $p$. Then, the function $\tau_{\text{NR}}^p(\cdot)$ has a local maximum at $x = \dfrac{1}{2 - 2^{-\frac{1}{p-1}}}$. Furthermore, its minimum value converges to $\frac{1}{4}$. See Figure 8.*

**Proof.** From (14), we know that $\tau_{\text{NR}}^p(x) = x^p - (2x - 1)_+^p$ where $p > 1$ and $p$ is an odd integer. Computing the first derivative of this function gives

$$\left(\tau_{\text{NR}}^p\right)'(x) = p\left(x^{(p-1)} - \left[\frac{(2x-1) + |2x-1|}{2}\right]^{(p-1)}\left(1 + \frac{2x-1}{|2x-1|}\right)\right).$$

To proceed, we discuss two cases:

Cases (i): If $x < \frac{1}{2}$, then $\left(\tau_{\text{NR}}^p\right)'(x) = px^{p-1} \geq 0$. Hence, $\tau_{\text{NR}}^p(x)$ is increasing on $(-\infty, \frac{1}{2})$, which indicates that it does not have local minimum or maximum value.

Cases (ii): If $x > \frac{1}{2}$, then $\left(\tau_{\text{NR}}^p\right)'(x) = p[x^{p-1} - 2(2x - 1)^{p-1}]$. It is verified that $a = \dfrac{1}{2 - 2^{-\frac{1}{p-1}}}$ is the only root of $p[x^{p-1} - 2(2x - 1)^{p-1}] = 0$ for $p > 1$. Moreover, we have that $\tau_{\text{NR}}^p(x)$ is decreasing (increasing) on the right (left) hand side of the point $a$. Hence, $a$ is a local

maximizer and the local maximum value is $\left[\dfrac{1}{2-2^{-\frac{1}{p-1}}}\right]^p - \left[2(\dfrac{1}{2-2^{-\frac{1}{p-1}}}) - 1\right]^p$. Furthermore,

the local maximum value $\left[\dfrac{1}{2-2^{-\frac{1}{p-1}}}\right]^p - \left[2(\dfrac{1}{2-2^{-\frac{1}{p-1}}}) - 1\right]^p$ converges to $\frac{1}{4}$ as $p \to \infty$. $\square$

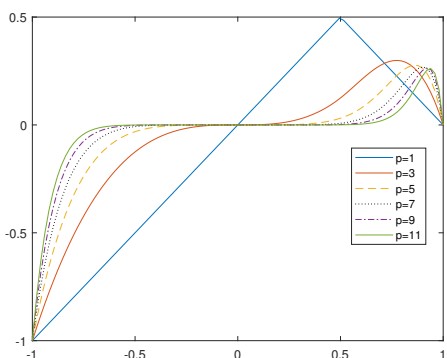

**Figure 8.** Local maximum of $\tau_{\mathrm{NR}}^p(x)$ with different values of $p$.

**Proposition 15.** *Let $\tau_{\mathrm{S-NR}}^p$ and $\sigma_{\mathrm{S-NR}}^p$ be defined as in (15) and (16), respectively. Then, for the odd integer $p$, the following hold. See Figure 9.*

(a) *The function $\tau_{\mathrm{S-NR}}^p(\cdot)$ has a local maximum at $x = 1 - \left( \dfrac{1}{2 - 2^{-\frac{1}{p-1}}} \right)$ and $\dfrac{1}{2 - 2^{-\frac{1}{p-1}}}$. Its local maximum value converges to $\frac{1}{4}$. Furthermore, it has a local minimum at $x = \frac{1}{2}$, which converges to 0.*

(b) *The function $\sigma_{\mathrm{S-NR}}^p(\cdot)$ has a local maximum at $x = \frac{1}{2}$ and its maximum value converges to 0. In addition, it has a local minimum at $x = 0$ and $x = 1$.*

**Proof.** (a) From (15), we know that

$$\tau_{\mathrm{S-NR}}^p(x) = \begin{cases} x^p - (2x - 1)^p, & \text{if } x > \frac{1}{2}, \\ (\frac{1}{2})^p, & \text{if } x = \frac{1}{2}, \\ (1 - x)^p - (1 - 2x)^p, & \text{if } x < \frac{1}{2}, \end{cases}$$

where $p$ is an odd integer. As $\tau_{\mathrm{S-NR}}^p(x)$ is symmetric at the point $x = \frac{1}{2}$, we consider the below two cases:

Cases (i): If $x > \frac{1}{2}$, according to Proposition 14, the local maximum point is $a = \dfrac{1}{2 - 2^{-\frac{1}{p-1}}}$ and

the maximum value is $\left[ \dfrac{1}{2 - 2^{-\frac{1}{p-1}}} \right]^p - [2(\dfrac{1}{2 - 2^{-\frac{1}{p-1}}}) - 1]^p$, which converges to $\frac{1}{4}$ as $p \to \infty$.

Cases (ii): If $x < \frac{1}{2}$, similar to Case (i), we obtain that $a = 1 - \left( \dfrac{1}{2 - 2^{-\frac{1}{p-1}}} \right)$ is a local

maximum point and the maximum value is $\left( \dfrac{1}{2 - 2^{-\frac{1}{p-1}}} \right)^p - \left( -1 + \dfrac{2}{2 - 2^{-\frac{1}{p-1}}} \right)^p$, which

converges to $\frac{1}{4}$ as $p \to \infty$.

Furthermore, because the function is increasing (decreasing) on the right (left) hand side of the point $a = \frac{1}{2}$, we can conclude $a = \frac{1}{2}$ is a local minimizer. Its the minimum value is $(\frac{1}{2})^p$, which converges to 0 when $p \to \infty$.

(b) From (16), we know that

$$\sigma_{\mathrm{S-NR}}^p(x) = \begin{cases} x^p(1 - x)^p - (2x - 1)^p(1 - x)^p, & \text{if } x > \frac{1}{2}, \\ (\frac{1}{2})^{2p}, & \text{if } x = \frac{1}{2}, \\ x^p(1 - x)^p - x^p(1 - 2x)^p, & \text{if } x < \frac{1}{2}, \end{cases}$$

where $p$ is an odd integer. Since $\sigma_{\mathrm{S-NR}}^p(x)$ is symmetric at the point $x = \frac{1}{2}$, we divide it into two cases:

Case (i): Suppose $x \geq \frac{2}{3}$, the first derivative is

$$\left(\sigma_{\text{S-NR}}^p\right)'(x) = -p(1-x)^{p-1}[x^p - (2x-1)^p] + (1-x)^p\left[p(x^{p-1} - 2(2x-1)^{p-1})\right].$$

Based on this, it is verified that $a = 1$ is a critical point. Because $\sigma_{\text{S-NR}}^p(x)$ is non-negative and $\sigma_{\text{S-NR}}^p(1) = 0$, we can conclude that 1 is a local minimum point and the value is 0.

Case (ii): Suppose $x \leq \frac{1}{3}$. Based on symmetry, the local minimum point is $a = 0$ and the value is 0.

Case (iii): Suppose $\frac{1}{3} < x < \frac{2}{3}$, we know that $\left(\sigma_{\text{S-NR}}^p\right)'(\frac{1}{2}) = 0$ and $\sigma_{\text{S-NR}}^p(x)$ is decreasing (increasing) on the right (left) side of the point $a = \frac{1}{2}$. Hence, we obtain that $a = \frac{1}{2}$ is a local maximizer and the maximum value is $(\frac{1}{2})^{2p}$ for $p \geq 3$. It clearly converges to 0 when $p \to \infty$. $\square$

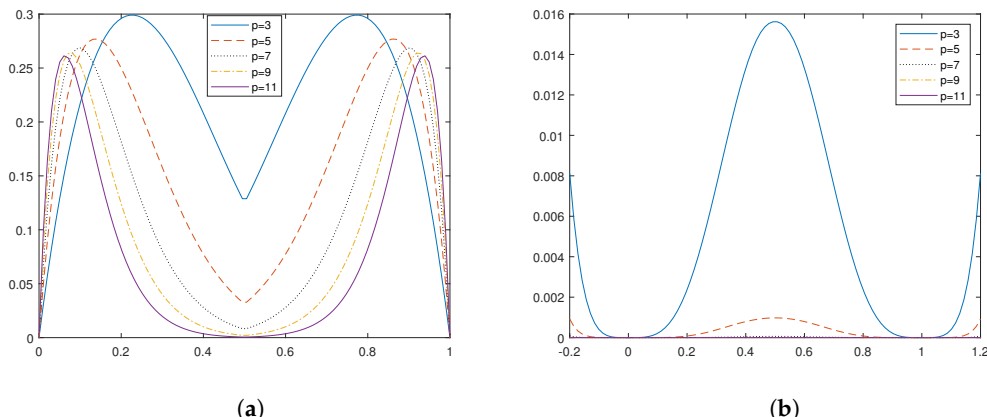

**Figure 9.** Graphs of $\tau_{\text{S-NR}}^p(x)$ and $\sigma_{\text{S-NR}}^p(x)$ with different values of $p$. (**a**) Local minimum and maximum of $\tau_{\text{S-NR}}^p(x)$ ; (**b**) Local minimum and maximum of $\sigma_{\text{S-NR}}^p(x)$.

Due to the fact that $\widetilde{\tau}_{\text{NR}}^p$, $\widetilde{\tau}_{\text{S-NR}}^p$, and $\widetilde{\sigma}_{\text{S-NR}}^p$ are continuous counterparts of $\tau_{\text{NR}}^p$, $\tau_{\text{S-NR}}^p$ and $\sigma_{\text{S-NR}}^p$, analogous to Propositions 14 and 15, their local maximums and minimums can be obtained. We omit the proof here.

**Proposition 16.** *Let $\widetilde{\tau}_{\text{NR}}^p(x)$, $\widetilde{\tau}_{\text{S-NR}}^p$ and $\widetilde{\sigma}_{\text{S-NR}}^p$ be defined as in (17), (18), and (19), respectively. Then, for $p > 1$, the following hold. See Figure 10.*

(a) *The function $\tau_{\text{NR}}^p(\cdot)$ has a local maximum at $x = \dfrac{1}{2 - 2^{-\frac{1}{p-1}}}$. Furthermore its minimum value converges to $\frac{1}{4}$.*

(b) *The function $\widetilde{\tau}_{\text{S-NR}}^p(\cdot)$ has a local maximum at $x = 1 - \left(\dfrac{1}{2 - 2^{-\frac{1}{p-1}}}\right)$ and $\dfrac{1}{2 - 2^{-\frac{1}{p-1}}}$ and its local maximum value converges to $\frac{1}{4}$. Furthermore, it has a local minimum at $x = \frac{1}{2}$ and converges to 0.*

(c) *The function $\widetilde{\sigma}_{\text{S-NR}}^p(\cdot)$ has a local maximum at $x = \frac{1}{2}$ and its local maximum value converges to 0. In addition, it has a local minimum at $x = 0$ and $x = 1$.*

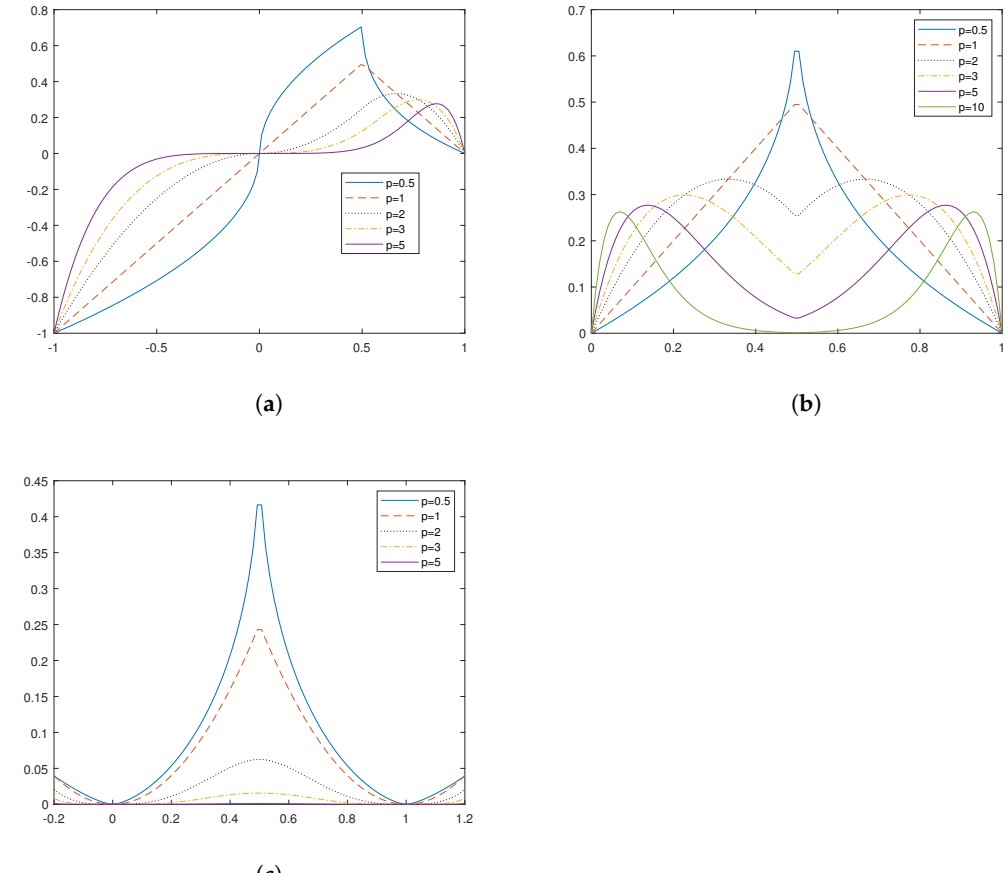

**Figure 10.** Graphs of $\widetilde{\tau}_{\mathrm{NR}}^{p}(x)$, $\widetilde{\tau}_{\mathrm{S-NR}}^{p}(x)$ and $\widetilde{\sigma}_{\mathrm{S-NR}}^{p}(x)$ with different values of $p$. (**a**) Local maximum of $\widetilde{\tau}_{\mathrm{NR}}^{p}(x)$; (**b**) Local minimum and maximum of $\widetilde{\tau}_{\mathrm{S-NR}}^{p}(x)$; (**c**) Local minimum and maximum of $\widetilde{\sigma}_{\mathrm{S-NR}}^{p}(x)$.

The local minimum for other $\tau_{f_i, g_i}(\cdot)$ is simple.

**Proposition 17.** *Let* $\tau_{f_i, g_i}$ *with* $i = 1, 2, 3$ *be defined as in* (20)–(28). *Then, the function* $\tau_{f_i, g_i}(\cdot)$ *has a local minimum at* $x = \frac{1}{2}$. *See Figure 11.*

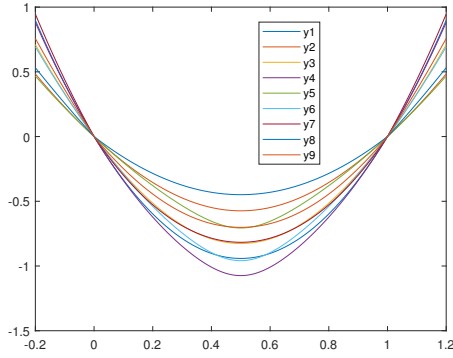

**Figure 11.** Local minimum of $\tau_{f_i, g_i}(x)$ for $i, j = 1, 2, 3$.

**Proof.** Because each $\tau_{f_i, g_i}(x)$ is nearly convex according to $x = \frac{1}{2}$ and $\tau_{f_i, g_i}(x)$ has a critical number at $x = \frac{1}{2}$, the local minimum at $x = \frac{1}{2}$ is confirmed and can be calculated easily. We only present the values here.

$$\tau_{f_1,g_1}\left(\frac{1}{2}\right) = \frac{1}{\sqrt{2}} - \sqrt{e}$$

$$\tau_{f_1,g_2}\left(\frac{1}{2}\right) = -\frac{1}{4} + \frac{1}{\sqrt{2}} - \frac{\sqrt{17}}{4}$$

$$\tau_{f_1,g_3}\left(\frac{1}{2}\right) = \frac{1}{\sqrt{2}} + \frac{1 - 4\sqrt{e}}{2 + \sqrt{e}}$$

$$\tau_{f_2,g_1}\left(\frac{1}{2}\right) = \frac{1}{2^{\frac{4}{5}}} - \sqrt{e}$$

$$\tau_{f_2,g_2}\left(\frac{1}{2}\right) = -\frac{1}{4} + \frac{1}{2^{\frac{4}{5}}} - \frac{\sqrt{17}}{4}$$

$$\tau_{f_2,g_3}\left(\frac{1}{2}\right) = \frac{1}{2^{\frac{4}{5}}} + \frac{1 - 4\sqrt{e}}{2 + \sqrt{e}}$$

$$\tau_{f_3,g_1}\left(\frac{1}{2}\right) = \ln(2\sqrt{e} - 1) - \sqrt{e}$$

$$\tau_{f_3,g_2}\left(\frac{1}{2}\right) = \ln(2\sqrt{e} - 1) - \frac{1}{4} - \frac{\sqrt{17}}{4}$$

$$\tau_{f_3,g_3}\left(\frac{1}{2}\right) = \ln(2\sqrt{e} - 1) + \frac{1 - 4\sqrt{e}}{2 + \sqrt{e}}$$

This completes the proof. □

## 6. Summary

To summarize, when comparing all the curves based on NCP functions, almost all of them are neither convex nor concave. Only the curve based on the Fischer–Burmeister function is convex due the fact that its corresponding NCP function is also convex. Nonetheless, we observe that some curves are convex whereas their corresponding NCP functions are not. For instance, the curve based on the discrete type of the Fischer–Burmeister function. This indicates that the convexity of the curves depends on the choice of vertical plane. In addition, when $p$ is perturbed, the interval of convexity will be shrunk or stretched. For the local minimum or maximum, when $p$ becomes very large, most of the minima and maxima converge. and the minima or maxima vary by the perturbation of $p$.

**Author Contributions:** Supervision, Y.-L.C. and J.-S.C.; writing—original draft, S.-W.L.; writing—review and editing, Y.-L.C. All authors have read and agreed to the published version of the manuscript.

**Funding:** National Taiwan Normal University and National Science and Technology Council, Taiwan.

**Institutional Review Board Statement:** Not applicable.

**Informed Consent Statement:** Not applicable.

**Data Availability Statement:** Not applicable.

**Conflicts of Interest:** The authors declare no conflict of interest.

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
