# Peer review of "Plane Section Curves on Surfaces of NCP Functions"

_axioms, doi:10.3390/axioms11100557_

Round 1
Reviewer 1 Report
The whole paper is well-organized and contains all of the components I would expect. The sections are well-developed and clearly explained. The paper gives the reader which is interested in this topic many results. Especially, the author drawn many beautiful figures to make the results more clearly.
In conclusion, the obtained results are meaningful, and they enrich its field of science. For more details, see the attachment file.

Author Response
We amend the typos and grammar mistakes according the refree's comments, and also change to the style of MDPI reference.
Thanks for the kindful comments from the refree.
Reviewer 2 Report
The authors study curves intersected by a vertical plane with surfaces based on certain NCP (nonlinear complementarity problem) functions. The results are interesting and the paper contains new ideas on local minimum and local maximum of the curves by using convexity and differentiability. The study on NCP functions is very useful to binary quadratic programming. The paper is well organized. But there are some errors in typos and grammar. For example,
(*) P.1(-9), `.’ should be replaced by `,’ and $F_1(x)$’ should be replaced by $F_1(x))$’
(*) p.1(-6); p.2(+5), What is a merit function?
(*) p.2(-3), odd integer
(30) found the
(39) as [6, Proposition 2.1 (b)]. (as \cite[Proposition 2.1 (b)]{6}.
(*) p.6(+8; +9; +10; +11; +14; +16; +17; +18; +19; +20); p.7(++1; +2; +3; +4; +5), add `.’
(106), Proposition 2(a)
(131) remove `. Therefore’
(271; 274), remove `entire’
(279), remove `on page 42’
(333), to Case (i)
(365), add `This completes the proof.’
(252; 264), remove `. Therefore’
(239), , it remains to
(*) p.16(+4), As
Ref [10] H. Tuy, ~ and Global
Ref [11] C.-H. Huang, J.-S. Chen
Ref [15] add Publication place
The authors have to read the final revised version before submitting the revised version.
Author Response
We are sorry for the unclearance. Defintion of merit function says the global minimizer of a merit function is a solution of NCP, and vise vesa. So solving NCP is equivalent to finding global minimizer of a merit function.
We add "of the NCP" after "a merit function" to emphasize the relation between both. We try not to put the definition here, since it is not the focus of this paper.
And we also amend all typos and grammar mistakes according to the refree.
Thanks for all kindful comments from the referee.
Reviewer 3 Report
The paper under review can be accepted for publication with slight modifications. In the first place, remove the citation [6] from the abstract. Second, capitalise "nonlinear complementary problem" in the first line of the Introduction and wherever it appears throughout the manuscript. Once you introduce the acronym NCP, there is no need to keep using "nonlinear complementary problem" throughout the paper. Just use the acronym. In page 1, in the definition of the vector-valued function there is a missing parenthesis in the first component.
It is clear the importance of NCP functions towards solving the NCP, however, the authors need to clarify the equivalence of solving NCP and solving the system of equations given by the vector-valued function constructed from the NCP function. What I mean is that it is clear that solving the system of equations implies solving NCP. How about the converse? That is, solving the NCP implies that there exists an NCP function whose associated system of equations is solved?
Finally, the authors mention that the study of NCP function based curves is very useful to binary quadratic programming, bu they do not provide an specific application.
Author Response
We remove the citation[6] from the abstact and capitalize "nonlinear complemetarity problem".
We are sorry for unclearance. The definition of NCP functions says \Phi(a,b)=0 is equivalent to a>=0, b>=0, <a,b>=0. So solving the system of equations implies solving NCP, and vice vesa.
Finally, we add the following words in the end of the instroduction:
We also have to point out that the study on these curves is very useful to binary quadratic programming. See [6] for the details.
We point out the reference[6] for applications to binary quadratic programming.
Thanks for your kindful comments.